# Generalization Bound for a Shallow Transformer Trained Using Gradient Descent

**Brian Mwigo**                                                                 *brian.mwigo@iitgn.ac.in*
*IIT Gandhinagar*

**Anirban Dasgupta**                                                           *anirbandg@iitgn.ac.in*
*IIT Gandhinagar*

**Reviewed on OpenReview:** *https://openreview.net/forum?id=t3iUeMOT8Z*

## Abstract

In this work, we establish a **norm-based generalization bound** for a *shallow Transformer model* trained via gradient descent under the *bounded-drift (lazy training)* regime, where model parameters remain close to their initialization throughout training. Our analysis proceeds in three stages: (a) we formally define a hypothesis class of Transformer models constrained to remain within a small neighborhood of their initialization; (b) we derive an **upper bound on the Rademacher complexity** of this class, quantifying its effective capacity; and (c) we establish an **upper bound on the empirical loss** achieved by gradient descent under suitable assumptions on model width, learning rate, and data structure. Combining these results, we obtain a **high-probability bound on the true loss** that decays sublinearly with the number of training samples $N$ and depends explicitly on model and data parameters. The resulting bound demonstrates that, in the lazy regime, wide and shallow Transformers generalize similarly to their linearized (NTK) counterparts. Empirical evaluations on both text and image datasets support the theoretical findings.

## 1 Introduction

Deep learning models have achieved remarkable success on language and vision tasks that were once considered intractable for neural networks. Transformer architectures, in particular, have played a central role in this progress, underpinning large-scale language models such as GPT-4 (Achiam et al., 2023), LLaMA (Touvron et al., 2023), and Gemini (Team et al., 2023). Beyond language, Vision Transformers (Dosovitskiy et al., 2020) have demonstrated competitive or superior performance on image classification and generation tasks. These empirical successes have fueled optimism about the capabilities of modern neural networks and, in some cases, speculation about the emergence of artificial general intelligence. Despite this progress, the theoretical understanding of Transformer models particularly their generalization behavior remains limited.

A central challenge in the theory of deep learning is to explain why modern neural network models generalize well. For Transformers, several recent works have developed generalization bounds by controlling the *generalization gap*, that is, the difference between the true loss and the empirical loss, $\mathcal{L}_{\mathcal{D}}(f) - \mathcal{L}_S(f)$, over a suitable hypothesis class $\mathcal{F}$ (Edelman et al., 2021; Trauger & Tewari, 2024; Fu et al., 2024). Such bounds are valuable for understanding capacity control and architectural inductive biases, but they require training to complete in order to evaluate the empirical loss $\mathcal{L}_S(f)$ before any conclusions about the true loss can be drawn.

An alternative line of work directly upper-bounds the *true loss* $\mathcal{L}_{\mathcal{D}}(f)$ by combining statistical complexity control with explicit analysis of optimization dynamics. This approach yields guarantees on generalization without requiring post hoc evaluation of the empirical loss. For fully connected networks, Arora et al. (2019b) established such a bound for overparameterized two-layer ReLU networks trained by gradient descent, leveraging convergence results in the neural tangent kernel (NTK) regime. This framework was also applied to

deep fully connected networks using Neural Tangent Random Features in Cao & Gu (2019). However, this optimization-aware true-loss perspective is less explored for Transformer architectures.

In this work, we take a step toward closing this gap by deriving a generalization bound for a class of Transformer models whose parameters remain close to their initialization throughout training. This bounded-drift assumption corresponds to the *lazy training regime*, which is commonly associated with highly overparameterized networks trained using small learning rates. Under this regime, training dynamics are approximately linearized around initialization, enabling tractable analysis while still capturing important aspects of overparameterized learning.

Our analysis proceeds in three steps. First, we formally define a hypothesis class of shallow Transformer models whose parameters remain within a fixed-radius neighborhood of their initialization. Second, we derive an upper bound on the Rademacher complexity of this class using covering-number arguments, thereby controlling its statistical capacity. Third, we leverage a recent global convergence theorem for shallow Transformers (Wu et al., 2024) to obtain an explicit upper bound on the empirical loss for all models in this class. Combining these ingredients yields a high-probability upper bound on the true loss.

*Specifically, our main contribution is an optimization-aware generalization bound that directly upper-bounds the true loss for a class of shallow Transformer models trained in the bounded-drift (lazy) regime. The resulting bound decreases sublinearly with the number of training samples $N$ and holds uniformly over model dimension $d_m$, under suitable overparameterization and data assumptions.*

## 2 Related Work

**Generalization bounds for neural networks.** Generalization in neural networks has been extensively studied using classical complexity measures such as VC dimension, Rademacher complexity, and covering numbers. Early norm- and margin-based bounds for fully connected networks were developed in Bartlett et al. (2017); Neyshabur et al. (2015; 2017), with later refinements incorporating spectral norms, path norms, and data-dependent quantities (Neyshabur et al., 2018; Pitas et al., 2018; Golowich et al., 2017; Li et al., 2018; Arora et al., 2018). PAC-Bayes and compression-based approaches further improved the tightness of these bounds and, in some cases, yielded non-vacuous guarantees for large networks (Zhou et al., 2018; Chen et al., 2019; Long & Sedghi, 2019). Despite this progress, classical complexity-based bounds are widely recognized to be quantitatively loose for modern overparameterized architectures, motivating alternative perspectives that incorporate optimization dynamics or data-dependent structure.

**Generalization bounds for Transformers.** Several works have specialized norm-based generalization analyses to Transformer architectures. Edelman et al. (2021) derive a generalization gap bound for Transformers that scales logarithmically with sequence length by exploiting bounded self-attention norms and sparse variable creation. This line was strengthened by Trauger & Tewari (2024), who obtain sequence-length-independent generalization gap bound. Beyond norm-based bounds, PAC-Bayes and algorithmic stability frameworks have been applied to Transformers and large language models. Compression-based PAC-Bayes analyses yield non-vacuous generalization bounds for large pretrained models by relating generalization to compressibility and posterior complexity (Lotfi et al., 2023). Stability-based approaches analyze sensitivity to data or task perturbations; for example, Li et al. (2023b) study in-context learning through multitask stability, while related stability analyses for attention and fine-tuning appear in Yao et al. (2025). Yang et al. (2025b) study generalization of Transformers for multi-step reasoning tasks.

**Optimization-aware bounds and NTK-based analyses.** A different line of work derives generalization guarantees by explicitly coupling optimization dynamics with statistical complexity control in overparameterized regimes. For fully connected networks, Arora et al. (2019b) establish a true-loss bound for two-layer ReLU networks trained by gradient descent using neural tangent kernel (NTK) theory, and Cao & Gu (2019) apply this approach to deep networks via neural tangent random features. More broadly, NTK-based analyses characterize convergence and generalization behavior of wide networks under lazy training dynamics (Jacot et al., 2018; Arora et al., 2019a). These works motivate optimization-aware analyses for more structured architectures such as Transformers.

**Training dynamics of Transformers.** Understanding the optimization and convergence behavior of Transformers has recently attracted significant attention. Wu et al. (2024) establish global convergence guarantees for shallow Transformers under suitable overparameterization and data assumptions, providing a Transformer analogue of NTK-style results for fully connected networks. Related works analyze convergence rates, representational dynamics, and optimization stability in attention-based architectures (Kohler & Krzyzak, 2023; Huang et al., 2024a; Shen et al., 2024; Gurevych et al., 2022; Ahn et al., 2023; Tarzanagh et al., 2023; Li et al., 2023a; Tian et al., 2023; Song et al., 2024; Huang et al., 2023; Chen & Li, 2024; Yang et al., 2024; Deora et al., 2023; Gao et al., 2024; Huang et al., 2024b; Jelassi et al., 2022; Wang et al., 2024; Yang et al., 2025a; Li et al., 2025; Yang et al., 2025b).

**Positioning of the present work.** Our work complements the above literature by providing, an *optimization-aware true-loss bound* for a Transformer architecture. Specifically, we combine (i) a Rademacher complexity bound for a bounded-drift Transformer hypothesis class with (ii) an explicit empirical-loss bound derived from a global convergence theorem for shallow Transformers (Wu et al., 2024). Unlike norm-based, PAC-Bayes, and stability-based approaches that bound $\mathcal{L}_{\mathcal{D}}(f) - \mathcal{L}_S(f)$, our analysis directly upper-bounds $\mathcal{L}_{\mathcal{D}}(f)$ under lazy-regime training dynamics. We emphasize that our results apply to single-layer Transformers under bounded drift and are therefore complementary i.e., not directly comparable to existing analyses of deep or non-bounded drift Transformer models.

## 3 Preliminaries

### 3.1 Problem Setup

#### 3.1.1 Training Examples

We are given $N$ training examples $S = \{(\boldsymbol{X}_n, y_n)\}_{n=1}^N$ where $\{\boldsymbol{X}_n\}_{n=1}^N \in \mathbb{R}^{N \times d_s \times d}$ are the instances and $\boldsymbol{y} \triangleq \{y_n\}_{n=1}^N \in \mathbb{R}^N$ are the labels. $d_s$ is the sequence length of the inputs and $d$ is the input dimension.

#### 3.1.2 Model

The model used in this work is a popular transformer encoder which is also used by Wu et al. (2024). Given an input $\boldsymbol{X} \in \mathbb{R}^{d_s \times d}$, we define each of the transformer layers.

***Self-attention layer***
The self-attention layer is defined as follows;

$$\boldsymbol{A}_1 \triangleq \sigma_s \left( \frac{(\boldsymbol{X}\boldsymbol{W}_Q^\top)(\boldsymbol{X}\boldsymbol{W}_K^\top)^\top}{\sqrt{d_m}} \right) (\boldsymbol{X}\boldsymbol{W}_V^\top)$$

where $\sigma_s$ is the row-wise softmax, $\boldsymbol{W}_Q, \boldsymbol{W}_K, \boldsymbol{W}_V \in \mathbb{R}^{d_m \times d}$ are the query, key and value matrices in the self-attention layer. $d_m$ is the model dimension. We shall be interested in the effect of the self-attention layer on each row $\boldsymbol{X}^{(i,:)}$ of the input $\boldsymbol{X}$ where $i \in [d_s]$. We therefore define $\beta_i$ as the $i$-th row of the softmax output;

$$\beta_i = \sigma_s \left( \frac{\boldsymbol{X}^{(i,:)}\boldsymbol{W}_Q^\top \boldsymbol{W}_K \boldsymbol{X}^\top}{\sqrt{d_m}} \right)^\top = \sigma_s \left( \frac{\boldsymbol{X}\boldsymbol{W}_K^\top \boldsymbol{W}_Q (\boldsymbol{X}^{(i,:)})^\top}{\sqrt{d_m}} \right)$$

We also define $\boldsymbol{z}_i$ as the final output of the self-attention layer for each row $\boldsymbol{X}^{(i,:)}$;

$$\boldsymbol{z}_i = (\boldsymbol{X}\boldsymbol{W}_V^\top)^\top \beta_i = \boldsymbol{W}_V \boldsymbol{X}^\top \sigma_s \left( \frac{\boldsymbol{X}\boldsymbol{W}_K^\top \boldsymbol{W}_Q (\boldsymbol{X}^{(i,:)})^\top}{\sqrt{d_m}} \right)$$

### *Feed -forward ReLU layer*

The layer with ReLU activation function is defined as follows;

$$\boldsymbol{A}_2 \triangleq \sigma_r(\boldsymbol{A}_1 \boldsymbol{W}_H)$$

where $\sigma_r$ is the ReLU activation function. For ease of calculations, $\boldsymbol{W}_H$ is set as $\boldsymbol{W}_H = \boldsymbol{I} \in \mathbb{R}^{d_m \times d_m}$ Once again, define $\boldsymbol{k}_i$ as the final output of the Feed -forward ReLU layer for each row $\boldsymbol{X}^{(i,:)}$;

$$\boldsymbol{k}_i = \sigma_r(\boldsymbol{z}_i) = \sigma_r \left( \boldsymbol{W}_V \boldsymbol{X}^\top \sigma_s \left( \frac{\boldsymbol{X} \boldsymbol{W}_K^\top \boldsymbol{W}_Q (\boldsymbol{X}^{(i,:)})^\top}{\sqrt{d_m}} \right) \right)$$

### *Average Pooling layer*

The pooling is applied column-wise to reduce sequence length dimension from $d_s$ to 1. This is done to ensure a scalar output from our transformer.

$$\boldsymbol{a}_3 \triangleq \varphi(\boldsymbol{A}_2)$$

where $\varphi$ represents the column-wise average pooling. We can also define $\boldsymbol{a}_3$ in terms of each $k_i$;

$$\boldsymbol{f}_{pre} = \frac{1}{d_s} \sum_{i=1}^{d_s} \boldsymbol{k}_i = \frac{1}{d_s} \sum_{i=1}^{d_s} \sigma_r \left( \boldsymbol{W}_V \boldsymbol{X}^\top \sigma_s \left( \frac{\boldsymbol{X} \boldsymbol{W}_K^\top \boldsymbol{W}_Q (\boldsymbol{X}^{(i,:)})^\top}{\sqrt{d_m}} \right) \right)$$

### *Output layer*

The final output layer is defined as follows;

$$f(\boldsymbol{X}) \triangleq \boldsymbol{w}_O^\top \boldsymbol{f}_{pre}$$

where $\boldsymbol{w}_O \in \mathbb{R}^{d_m}$ is the weight vector in the output layer. We can as well define the final model output $f(\boldsymbol{X})$ in terms of each row $\boldsymbol{X}^{(i,:)}$ of the input $\boldsymbol{X}$;

$$f(\boldsymbol{X}) = \frac{1}{d_s} \boldsymbol{w}_O^\top \sum_{i=1}^{d_s} \sigma_r \left( \boldsymbol{W}_V \boldsymbol{X}^\top \sigma_s \left( \frac{\boldsymbol{X} \boldsymbol{W}_K^\top \boldsymbol{W}_Q (\boldsymbol{X}^{(i,:)})^\top}{\sqrt{d_m}} \right) \right)$$

Define $\boldsymbol{\theta}$ as a vector representing the union of all parameters of the transformer model as shown below;

$$\boldsymbol{\theta} = \{\boldsymbol{W}_Q, \boldsymbol{W}_K, \boldsymbol{W}_V, \boldsymbol{w}_O\}$$

When we pass a single input $\boldsymbol{X} \in \mathbb{R}^{d_s \times d}$ to the model, the output is given as $f(\boldsymbol{X}) \in \mathbb{R}$. When we give all inputs to the model as a batch $\{\boldsymbol{X}_n\}_{n=1}^N \in \mathbb{R}^{N \times d_s \times d}$, the output of the model will be $\boldsymbol{f} \triangleq \{f(\boldsymbol{X}_n)\}_{n=1}^N \in \mathbb{R}^N$ and output of the last hidden layer will be $\boldsymbol{F}_{pre} \triangleq \{\boldsymbol{f}_{pre}(\boldsymbol{X_n})\}_{n=1}^N \in \mathbb{R}^{N \times d_m}$.

### 3.1.3 Initialization

Similar to Wu et al. (2024) we use the LeCun initialization described below. The parameters $\boldsymbol{W}_Q, \boldsymbol{W}_K, \boldsymbol{W}_V$ are initialized as $\boldsymbol{W}_Q^{(ij)} \sim \mathcal{N}(0, \frac{1}{d})$, $\boldsymbol{W}_K^{(ij)} \sim \mathcal{N}(0, \frac{1}{d})$, $\boldsymbol{W}_V^{(ij)} \sim \mathcal{N}(0, \frac{1}{d})$ for $i \in [d_m]$ and $j \in [d]$ while $\boldsymbol{w}_O^{(i)}$ is initialized as $\boldsymbol{w}_O^{(i)} \sim \mathcal{N}(0, \frac{1}{d_m})$ for $i \in [d_m]$.

### 3.1.4 Empirical Loss

We consider any loss function $\ell(f(\boldsymbol{X}_n), y_n)$ which is 1-Lipschitz in the first argument;

$$\mathcal{L}_S(f) = \frac{1}{N} \sum_{n=1}^N \ell(f(\boldsymbol{X}_n), y_n)$$

This empirical loss is to be optimized using Gradient Descent algorithm shown below;

**Input:** data $(\boldsymbol{X}_n, y_n)_{n=1}^N$, step size $\gamma$
Initialize weights as follows: $\boldsymbol{\theta}^0 := \{\boldsymbol{W}_Q^0, \boldsymbol{W}_K^0, \boldsymbol{W}_V^0, \boldsymbol{w}_O^0\}$
***for*** $t = 0$ ***to*** $t' - 1$ ***do***
$\boldsymbol{W}_Q^{t+1} = \boldsymbol{W}_Q^t - \gamma \cdot \nabla_{\boldsymbol{W}_Q} \ell(\boldsymbol{\theta}^t)$
$\boldsymbol{W}_K^{t+1} = \boldsymbol{W}_K^t - \gamma \cdot \nabla_{\boldsymbol{W}_K} \ell(\boldsymbol{\theta}^t)$
$\boldsymbol{W}_V^{t+1} = \boldsymbol{W}_V^t - \gamma \cdot \nabla_{\boldsymbol{W}_V} \ell(\boldsymbol{\theta}^t)$
$\boldsymbol{w}_O^{t+1} = \boldsymbol{w}_O^t - \gamma \cdot \nabla_{\boldsymbol{w}_O} \ell(\boldsymbol{\theta}^t)$
***end for***
**Output:** the model based on $\boldsymbol{\theta}^{t'}$.

### 3.1.5 True Loss

We are interested in upper bounding the true loss defined as follows;

$$\mathcal{L}_\mathcal{D}(f) = \mathbb{E}_{(\boldsymbol{X}, y) \sim \mathcal{D}}[\ell(f(\boldsymbol{X}), y)]$$

### 3.2 Rademacher complexity

The theorem of Rademacher complexity is widely used to compute generalization bounds for machine learning models. As per Mohri et al. (2012) theorem 3.1 and Arora et al. (2019b) theorem B.1, suppose that the loss function $\ell(\cdot, \cdot)$ is bounded in $[0, c]$ and is $\rho$-Lipschitz in the first argument. Then with probability at least $1 - \delta$ over the sample $S = \{(\boldsymbol{X}_n, y_n)\}_{n=1}^N$ of size $N$:

$$\sup_{f \in \mathcal{F}}\{\mathcal{L}_\mathcal{D}(f) - \mathcal{L}_S(f)\} \le 2\rho\mathcal{R}_S(\mathcal{F}) + 3c\sqrt{\frac{\log(2/\delta)}{2N}}$$

where $\mathcal{L}_\mathcal{D}(f)$ is the true loss, $\mathcal{L}_S(f)$ is the empirical loss and $\mathcal{R}_S(\mathcal{F})$ is the empirical Rademacher complexity of a function class $\mathcal{F}$ for samples $S = \{(\boldsymbol{X}_n, y_n)\}_{n=1}^N$ of size $N$ defined as follows;

$$\mathcal{R}_S(\mathcal{F}) = \frac{1}{N}\mathbb{E}_{\epsilon \sim \text{unif}(\{1, -1\})}\left[\sup_{f \in \mathcal{F}} \sum_{n=1}^N \epsilon_n f(\boldsymbol{X}_n)\right]$$

In order to construct our generalization bound, we shall upper bound both the Rademacher complexity $\mathcal{R}_S(\mathcal{F})$ and the training loss $\mathcal{L}_S(f)$ for all $f \in \mathcal{F}$.

### 3.3 Covering number bound

For a given class $\mathcal{F}$, the covering number $\mathcal{N}_\infty(\mathcal{F}; \epsilon; \{\boldsymbol{X}_n\}_{n=1}^N; \|\cdot\|_2)$ is the smallest size of a collection (a cover) $\mathcal{C} \subset \mathcal{F}$ such that $\forall f \in \mathcal{F}, \exists \hat{f} \in \mathcal{C}$ satisfying $\max_n \|f(\boldsymbol{X}_n) - \hat{f}(\boldsymbol{X}_n)\|_2 \le \epsilon$.

The Rademacher complexity of the class $\mathcal{F}$ with respect to samples $S = \{(\boldsymbol{X}_n, y_n)\}_{n=1}^N$ can be upper bounded using the covering number of $\mathcal{F}$ (Edelman et al., 2021);

$$\mathcal{R}_S(\mathcal{F}) \le c \cdot \inf_{\delta \ge 0}\left(\delta + \int_\delta^A \sqrt{\frac{\log\mathcal{N}_\infty(\mathcal{F}; \epsilon; \{\boldsymbol{X}_n\}_{n=1}^N; \|\cdot\|_2)}{N}}d\epsilon\right)$$

for some constant $c > 0$ and $|f| \le A$ for all $f \in \mathcal{F}$.

# 4 Results

In this section, we develop a theoretical framework to analyze the generalization properties of Transformer models whose parameters remain close to their initialization during training. We begin by formally defining a class of models that satisfy this bounded-drift property, which corresponds to the *lazy training regime.* We then derive an upper bound on the Rademacher complexity of this class, followed by an upper bound on the empirical loss using convergence guarantees under gradient descent. Combining these results, we present our main theorem that establishes a high-probability bound on the true loss. Finally, we discuss the scope and limitations of our findings in light of existing results, and conclude with key insights and directions for future work.

For ease of proof, and without loss of generality, let us set the input feature dimension $d$ to be equal to the model dimension $d_m$ i.e., $d = d_m$.

## 4.1 Defining a class of Transformer models whose weights stay close to their initialization

To rigorously analyze the generalization behavior of Transformers, we first need to formalize the notion of models whose parameters remain close to their initialization throughout training. This assumption (often referred to as the *bounded-drift assumption*) characterizes the lazy training regime, where model updates are small, and the network operates in a nearly linear regime around initialization. In this subsection, we define the parameter space and construct a hypothesis class of Transformer models confined within a ball of radius $R$ centered at the initialization point. This setup enables the derivation of subsequent complexity and loss bounds under analytically tractable conditions.

Recall that we defined $\boldsymbol{\theta}$ as a vector representing the union of all parameters of the transformer model as shown below;

$$\boldsymbol{\theta} = \{\boldsymbol{W}_Q, \boldsymbol{W}_K, \boldsymbol{W}_V, \boldsymbol{w}_O\}$$

The squared $\ell_2$-norm of the parameter vector can be expressed as the sum of the squared Frobenius norms (for matrices) and squared $\ell_2$-norms (for vectors);

$$\|\boldsymbol{\theta}\|_2^2 = \|\boldsymbol{W}_Q\|_F^2 + \|\boldsymbol{W}_K\|_F^2 + \|\boldsymbol{W}_V\|_F^2 + \|\boldsymbol{w}_O\|_2^2$$

We can therefore say that for all training steps $t > 0$;

$$\begin{aligned} \|\boldsymbol{\theta}^{t+1} - \boldsymbol{\theta}^0\|_2^2 &= \|\boldsymbol{W}_Q^{t+1} - \boldsymbol{W}_Q^0\|_F^2 + \|\boldsymbol{W}_K^{t+1} - \boldsymbol{W}_K^0\|_F^2 + \|\boldsymbol{W}_V^{t+1} - \boldsymbol{W}_V^0\|_F^2 + \|\boldsymbol{w}_O^{t+1} - \boldsymbol{w}_O^0\|_2^2 \\ &\leq R_Q^2 + R_K^2 + R_V^2 + R_O^2 \end{aligned}$$

where $\|\boldsymbol{W}_Q^{t+1} - \boldsymbol{W}_Q^0\|_F \leq R_Q, \|\boldsymbol{W}_K^{t+1} - \boldsymbol{W}_K^0\|_F \leq R_K, \|\boldsymbol{W}_V^{t+1} - \boldsymbol{W}_V^0\|_F \leq R_V, \|\boldsymbol{w}_O^{t+1} - \boldsymbol{w}_O^0\|_2 \leq R_O$ for some positive constants $R_O, R_V, R_Q, R_K$

Setting $R = \sqrt{R_Q^2 + R_K^2 + R_V^2 + R_O^2}$ gives $\|\boldsymbol{\theta}^{t+1} - \boldsymbol{\theta}^0\|_2 \leq R$. We then define our hypothesis class $\mathcal{F}_R^{\boldsymbol{\theta}^0}$ comprised of the transformer models whose parameters $\boldsymbol{\theta}$ stay in a ball close to $\boldsymbol{\theta}^0$ for all training steps $t > 0$;

$$\mathcal{F}_R^{\boldsymbol{\theta}^0} = \left\{ f_{\boldsymbol{\theta}}(\boldsymbol{X}_n) : \forall t > 0, \|\boldsymbol{\theta}^{t+1} - \boldsymbol{\theta}^0\|_2 \leq R \right\}$$

## 4.2 Upper bounding the Rademacher complexity

To establish a generalization bound, we must first control the capacity of the hypothesis class of models under consideration. The Rademacher complexity provides a data-dependent measure of this capacity,

quantifying how well the model class can fit random noise. In this subsection, we derive an upper bound on the Rademacher complexity of the bounded-drift Transformer class defined above. Our result shows that under reasonable assumptions on the input features and parameter norms, the Rademacher complexity scales as $\mathcal{O}\left(\sqrt{\frac{P}{N}}\log(A\sqrt{\frac{N}{P}})\right)$, indicating that generalization improves with an increasing number of samples and controlled parameter magnitudes. This bound parallels similar results for shallow transformer and provides the foundation for our overall generalization analysis.

The following lemma gives an upper bound on the Rademacher complexity of our class of transformer models i.e., an upper bound on $\mathcal{R}_S(\mathcal{F}_R^{\boldsymbol{\theta}^0})$.

**Lemma 1.** *Suppose that we have* $\eta_V = \|\boldsymbol{W}_V^0\|_F + R_V, \eta_O = \|\boldsymbol{w}_O^0\|_2 + R_O, \eta_K = \|\boldsymbol{W}_K^0\|_F + R_K, \eta_Q = \|\boldsymbol{W}_Q^0\|_F + R_Q$ *where* $R_O, R_V, R_K, R_Q$ *remain as defined above. Also assume that the inputs have full rank and are bounded as* $\|\boldsymbol{X}_n\|_F \leq \sqrt{d_s}R_X$ *for all* $n \in [N]$ *where* $R_X$ *is some positive constant. The empirical Rademacher complexity of the class of Transformer models* $\mathcal{F}_R^{\boldsymbol{\theta}^0} = \left\{f_{\boldsymbol{\theta}}(\boldsymbol{X}_n) : \forall t > 0, \|\boldsymbol{\theta}^{t+1} - \boldsymbol{\theta}^0\|_2 \leq R\right\}$ *given* $\boldsymbol{\theta} = \{\boldsymbol{W}_Q, \boldsymbol{W}_K, \boldsymbol{W}_V, \boldsymbol{w}_O\}$ *can be upper bounded as follows*

$$\mathcal{R}_S(\mathcal{F}_R^{\boldsymbol{\theta}^0}) \lesssim \mathcal{O}\left(\frac{1}{N}\sqrt{\frac{P}{N}}\left(1 + \log\left(A\sqrt{\frac{N}{P}}\right)\right)\right)$$

*where* $\lesssim$ *hides logarithmic dependencies on quantities besides* $N$ *and* $d_s$, $A = \eta_O\eta_V(\sqrt{d_s}R_X)$ *and* $P = (\sqrt{d_s}R_X)^2\left(\left(\sqrt{d_m}\eta_V\right)^{\frac{2}{3}} + \left(\sqrt{d_m}\eta_K\eta_Q\eta_V\right)^{\frac{2}{3}}\right)^3\log(Nd_s)$

*Proof.* Define the following quantities for simplicity $\eta_V = \|\boldsymbol{W}_V^0\|_F + R_V, \eta_O = \|\boldsymbol{w}_O^0\|_2 + R_O, \eta_K = \|\boldsymbol{W}_K^0\|_F + R_K, \eta_Q = \|\boldsymbol{W}_Q^0\|_F + R_Q$ where $R_O, R_V, R_K, R_Q$ remain as defined above in section 4.1.

Our class of interest in section 4.1 was $\mathcal{F}_R^{\boldsymbol{\theta}^0} = \left\{f_{\boldsymbol{\theta}}(\boldsymbol{X}_n) : \|\boldsymbol{\theta}^{t+1} - \boldsymbol{\theta}^0\|_2 \leq R\right\}$ and we want to compute upper bound on the empirical Rademacher complexity $\mathcal{R}_S(\mathcal{F}_R^{\boldsymbol{\theta}^0})$ which is given as follows:

$$\mathcal{R}_S(\mathcal{F}_R^{\boldsymbol{\theta}^0}) = \frac{1}{N}\mathbb{E}_{\epsilon\sim\text{unif}(\{-1,1\})}\left[\sup_{\substack{\boldsymbol{w}_O,\boldsymbol{W}_K^\top\boldsymbol{W}_Q,\boldsymbol{W}_V:\\ \|\boldsymbol{w}_O\|_2\leq\eta_O\\ \|\boldsymbol{W}_V\|_F\leq\eta_V\\ \left\|\frac{\boldsymbol{W}_K^\top\boldsymbol{W}_Q}{\sqrt{d_m}}\right\|_F\leq\frac{\eta_K\eta_Q}{\sqrt{d_m}}}}\sum_{n=1}^N\epsilon_n\frac{1}{d_s}\boldsymbol{w}_O^\top\sum_{i=1}^{d_s}\sigma_r\left(\boldsymbol{W}_V\boldsymbol{X}_n^\top\sigma_s\left(\frac{\boldsymbol{X}_n\boldsymbol{W}_K^\top\boldsymbol{W}_Q(\boldsymbol{X}_n^{(i,:)})^\top}{\sqrt{d_m}}\right)\right)\right]$$

$$= \frac{1}{Nd_s}\mathbb{E}_{\epsilon\sim\text{unif}(\{-1,1\})}\left[\sup_{\substack{\boldsymbol{w}_O,\boldsymbol{W}_K^\top\boldsymbol{W}_Q,\boldsymbol{W}_V:\\ \|\boldsymbol{w}_O\|_2\leq\eta_O\\ \|\boldsymbol{W}_V\|_F\leq\eta_V\\ \left\|\frac{\boldsymbol{W}_K^\top\boldsymbol{W}_Q}{\sqrt{d_m}}\right\|_F\leq\frac{\eta_K\eta_Q}{\sqrt{d_m}}}}\sum_{n=1}^N\epsilon_n\boldsymbol{w}_O^\top\sum_{i=1}^{d_s}\sigma_r\left(\boldsymbol{W}_V\boldsymbol{X}_n^\top\sigma_s\left(\frac{\boldsymbol{X}_n\boldsymbol{W}_K^\top\boldsymbol{W}_Q(\boldsymbol{X}_n^{(i,:)})^\top}{\sqrt{d_m}}\right)\right)\right]$$

Applying subadditivity of the supremum:

$$
\mathcal{R}_S(\mathcal{F}_R^{\boldsymbol{\theta}^0}) \leq \frac{1}{Nd_s} \sum_{i=1}^{d_s} \mathbb{E}_{\epsilon \sim \mathrm{unif}(\{-1,1\})} \left[ \sup_{\substack{\boldsymbol{w}_O, \boldsymbol{W}_K^\top \boldsymbol{W}_Q, \boldsymbol{W}_V: \\ \|\boldsymbol{w}_O\|_2 \leq \eta_O \\ \|\boldsymbol{W}_V\|_F \leq \eta_V \\ \left\| \frac{\boldsymbol{W}_K^\top \boldsymbol{W}_Q}{\sqrt{d_m}} \right\|_F \leq \frac{\eta_K \eta_Q}{\sqrt{d_m}}}} \sum_{n=1}^{N} \epsilon_n \, \boldsymbol{w}_O^\top \, \sigma_r \left( \boldsymbol{W}_V \boldsymbol{X}_n^\top \sigma_s \left( \frac{\boldsymbol{X}_n \boldsymbol{W}_K^\top \boldsymbol{W}_Q (\boldsymbol{X}_n^{(i,:)})^\top}{\sqrt{d_m}} \right) \right) \right]
$$

$$
= d_s \cdot \frac{1}{Nd_s} \mathbb{E}_{\epsilon \sim \mathrm{unif}(\{-1,1\})} \left[ \sup_{\substack{\boldsymbol{w}_O, \boldsymbol{W}_K^\top \boldsymbol{W}_Q, \boldsymbol{W}_V: \\ \|\boldsymbol{w}_O\|_2 \leq \eta_O \\ \|\boldsymbol{W}_V\|_F \leq \eta_V \\ \left\| \frac{\boldsymbol{W}_K^\top \boldsymbol{W}_Q}{\sqrt{d_m}} \right\|_F \leq \frac{\eta_K \eta_Q}{\sqrt{d_m}}}} \sum_{n=1}^{N} \epsilon_n \, \boldsymbol{w}_O^\top \, \sigma_r \left( \boldsymbol{W}_V \boldsymbol{X}_n^\top \sigma_s \left( \frac{\boldsymbol{X}_n \boldsymbol{W}_K^\top \boldsymbol{W}_Q (\boldsymbol{X}_n^{(i,:)})^\top}{\sqrt{d_m}} \right) \right) \right]
$$

$$
= \underbrace{\frac{1}{N} \mathbb{E}_{\epsilon \sim \mathrm{unif}(\{-1,1\})} \left[ \sup_{\substack{\boldsymbol{w}_O, \boldsymbol{W}_K^\top \boldsymbol{W}_Q, \boldsymbol{W}_V: \\ \|\boldsymbol{w}_O\|_2 \leq \eta_O \\ \|\boldsymbol{W}_V\|_F \leq \eta_V \\ \left\| \frac{\boldsymbol{W}_K^\top \boldsymbol{W}_Q}{\sqrt{d_m}} \right\|_F \leq \frac{\eta_K \eta_Q}{\sqrt{d_m}}}} \sum_{n=1}^{N} \epsilon_n \, \boldsymbol{w}_O^\top \, \sigma_r \left( \boldsymbol{W}_V \boldsymbol{X}_n^\top \sigma_s \left( \frac{\boldsymbol{X}_n \boldsymbol{W}_K^\top \boldsymbol{W}_Q (\boldsymbol{X}_n^{(i,:)})^\top}{\sqrt{d_m}} \right) \right) \right]}_{\triangleq \, \mathcal{R}_S(\mathcal{G}_R^{\boldsymbol{\theta}^0})}
$$

for any fixed $i \in [d_s]$. Hence,

$$
\mathcal{R}_S(\mathcal{F}_R^{\boldsymbol{\theta}^0}) \ \leq \ \mathcal{R}_S(\mathcal{G}_R^{\boldsymbol{\theta}^0})
$$

where $\mathcal{G}_R^{\boldsymbol{\theta}^0}$ is defined as follows

$$
\mathcal{G}_R^{\boldsymbol{\theta}^0} := \left\{ (\boldsymbol{X}^{(i,:)})^\top \mapsto \boldsymbol{w}_O^\top \sigma_r \left( \boldsymbol{W}_V \boldsymbol{X}_n^\top \sigma_s \left( \frac{\boldsymbol{X}_n \boldsymbol{W}_K^\top \boldsymbol{W}_Q (\boldsymbol{X}_n^{(i,:)})^\top}{\sqrt{d_m}} \right) \right) : \ \|\boldsymbol{w}_O\|_2 \leq \eta_O, \ \|\boldsymbol{W}_V\|_F \leq \eta_V, \ \left\| \frac{\boldsymbol{W}_K^\top \boldsymbol{W}_Q}{\sqrt{d_m}} \right\|_F \leq \frac{\eta_K \eta_Q}{\sqrt{d_m}} \right\}.
$$

The following lemma gives an upper bound on $\mathcal{R}_S(\mathcal{G}_R^{\boldsymbol{\theta}^0})$. Its proof can be found in the appendix section;

**Lemma 2.** *For any fixed $\epsilon > 0$ and $\boldsymbol{X}_1, \ldots, \boldsymbol{X}_N \in \mathbb{R}^{d_s \times d}$ such that $\|\boldsymbol{X}_n\|_F \leq \sqrt{d_s} R_X$ for all $n \in [N]$, the Rademacher complexity of $\mathcal{G}_R^{\boldsymbol{\theta}^0}$ satisfies the bound given below;*

$$
\mathcal{R}_S(\mathcal{G}_R^{\boldsymbol{\theta}^0}) \lesssim c \sqrt{\frac{P}{N}} \left( 1 + \log \left( A \sqrt{\frac{N}{P}} \right) \right)
$$

*where $\lesssim$ hides logarithmic dependencies on quantities besides $N$ and $d_s$, $A = \eta_O \eta_V (\sqrt{d_s} R_X)$ and $P = (\sqrt{d_s} R_X)^2 \left( \left( \sqrt{d_m} \eta_V \right)^{\frac{2}{3}} + \left( \sqrt{d_m} \eta_K \eta_Q \eta_V \right)^{\frac{2}{3}} \right)^3 \log(Nd_s)$.*

Finally, the upper bound on the Rademacher complexity $\mathcal{R}_S(\mathcal{F}_R^{\boldsymbol{\theta}^0})$ can be given as;

$$
\begin{aligned}
\mathcal{R}_S(\mathcal{F}_R^{\boldsymbol{\theta}^0}) &\leq \mathcal{R}_S(\mathcal{G}_R^{\boldsymbol{\theta}^0}) \\
&\lesssim \sqrt{\frac{P}{N}}\left(1 + \log\left(A\sqrt{\frac{N}{P}}\right)\right) \\
&\lesssim \mathcal{O}\left(\sqrt{\frac{P}{N}}\left(1 + \log\left(A\sqrt{\frac{N}{P}}\right)\right)\right)
\end{aligned}
$$

This completes the proof. $\qquad\square$

## 4.3 Upper bounding the empirical loss

Having bounded the complexity of our hypothesis class, we next analyze the empirical loss achieved by gradient descent under the bounded-drift condition. This subsection establishes that, given suitable conditions on model width, learning rate, and data structure, the empirical loss decays exponentially during training. Using results from convergence analyses of Transformers in the lazy regime, we derive an explicit upper bound on the empirical loss as a function of key quantities such as $\alpha$, $\rho$, and $\eta_O$. This provides a quantitative connection between network conditioning, data complexity, and training behavior, ensuring that even under restricted parameter updates, the model achieves low empirical loss with high probability.

Define $\alpha$ as the minimum singular value of $\boldsymbol{F}_{\text{pre}}^0$, i.e., $\alpha \triangleq \sigma_{\min}(\boldsymbol{F}_{\text{pre}}^0)$ and also define $\Phi(\boldsymbol{\theta})$ as follows;

$$
\Phi(\boldsymbol{\theta}) = \frac{1}{2}\|\boldsymbol{f}(\boldsymbol{\theta}) - \boldsymbol{y}\|_2^2
$$

We now state the following assumption about the input data matrix $\boldsymbol{X}$;

**Assumption 3.** *Assume that the input data has full row rank and is bounded as $\|\boldsymbol{X}\|_F \leq \sqrt{d_s}R_X$ with some positive constant $R_X$. Furthermore, For any data pair $(\boldsymbol{X}_n, \boldsymbol{X}_{n'})$, with $n \neq n'$ and $n, n' \in [N]$, then we assume that;*

$$
\mathbb{P}(|\langle \boldsymbol{X}_n^\top \boldsymbol{X}_n, \boldsymbol{X}_{n'}^\top \boldsymbol{X}_{n'}\rangle| \geq t) \leq \exp(-t^{\hat{c}})
$$

*with some constant $\hat{c} > 0$*

The lemma below gives an upper bound on the empirical loss for all training steps $t > 0$.

**Lemma 4.** *Suppose that we have $\eta_V = \|\boldsymbol{W}_V^0\|_F + R_V, \eta_O = \|\boldsymbol{w}_O^0\|_2 + R_O, \eta_K = \|\boldsymbol{W}_K^0\|_F + R_K, \eta_Q = \|\boldsymbol{W}_Q^0\|_F + R_Q, \xi_Q = \|\boldsymbol{W}_Q^0\|_2 + R_Q, \xi_K = \|\boldsymbol{W}_K^0\|_2 + R_K, \xi_V = \|\boldsymbol{W}_V^0\|_2 + R_V$ where $R_O, R_V, R_K, R_Q$ remain as defined earlier. Under assumption 3, if $d_m \geq \widetilde{\Omega}(N^3)$, $\alpha^2 \geq 8\rho M\sqrt{2\Phi(\boldsymbol{\theta}^0)}$, $\alpha^3 \geq (32\rho^2 z\sqrt{2\Phi(\boldsymbol{\theta}^0)})/\eta_O$ and $\ell(\boldsymbol{\theta})$ is any loss function which is 1-Lipschitz in the first argument, then with probability at least $1 - 8e^{-d_m/2} - \delta - \exp(-\Omega((N-1)^{-\hat{c}}d_s^{-1}))$, for proper $\delta$, when training using GD with small step size $\gamma \leq 1/k$ where $k$ is a constant depending on $(\xi_Q, \xi_K, \xi_V, \eta_O, \Phi(\boldsymbol{\theta}^0), \rho, d_m^{-1/2})$, the empirical loss can be bounded as follows for all $t > 0$;*

$$
\mathcal{L}_S(f_{\boldsymbol{\theta}^t}) \leq \min\left(\frac{\alpha^2}{8\rho\hat{M}\sqrt{N}}, \frac{\alpha^3\eta_O}{32\rho^2\hat{z}\sqrt{N}}\right)
$$

*where $\widetilde{\Omega}$ omits the logarithmic factor and the other quantities are defined as follows; $\rho \triangleq N^{1/2}d_s^{3/2}R_X$, $z \triangleq \eta_O^2(1 + (4/d_m)R_X^4 d_s^2\xi_V^2(\xi_Q^2 + \xi_K^2))$, $\hat{z} \triangleq \eta_O^2(1 + (4/d_m)R_X^4 d_s^2\eta_V^2(\eta_Q^2 + \eta_K^2))$, $M = \max(\xi_V R_O^{-1}, \eta_O R_V^{-1}, (2/\sqrt{d_m})R_X^2 d_s\xi_K\xi_V\eta_O R_Q^{-1}, 2/\sqrt{d_m})R_X^2 d_s\xi_Q\xi_V\eta_O R_K^{-1})$, $\hat{M} = \max(\eta_V R_O^{-1}, \eta_O R_V^{-1}, (2/\sqrt{d_m})R_X^2 d_s\eta_K\eta_V\eta_O R_Q^{-1}, (2/\sqrt{d_m})R_X^2 d_s\eta_Q\eta_V\eta_O R_K^{-1})$.*

*Proof.* For the purpose of simplification, define the following quantities at initialization;

$$\xi_Q \triangleq \|\boldsymbol{W}_Q^0\|_2 + R_Q \leq \|\boldsymbol{W}_Q^0\|_F + R_Q \triangleq \eta_Q$$

$$\xi_K \triangleq \|\boldsymbol{W}_K^0\|_2 + R_K \leq \|\boldsymbol{W}_K^0\|_F + R_K \triangleq \eta_K$$

$$\xi_V \triangleq \|\boldsymbol{W}_V^0\|_2 + R_V \leq \|\boldsymbol{W}_V^0\|_F + R_V \triangleq \eta_V$$

$$\eta_O \triangleq \|\boldsymbol{w}_O^0\|_2 + R_O$$

where $R_Q, R_K, R_V, R_O$ are as defined before. As mentioned earlier, $\alpha$ is the minimum singular value of $\boldsymbol{F}_{\text{pre}}^0$, i.e., $\alpha \triangleq \sigma_{\min}(\boldsymbol{F}_{\text{pre}}^0)$ and $\Phi(\boldsymbol{\theta})$ is given as $\Phi(\boldsymbol{\theta}) = \frac{1}{2}\|\boldsymbol{f}(\boldsymbol{\theta}) - \boldsymbol{y}\|_2^2$.

According to Wu et al. (2024) theorem 1, under assumption 3, if $d_m \geq \widetilde{\Omega}(N^3)$, $\alpha^2 \geq 8\rho M \sqrt{2\Phi(\boldsymbol{\theta}^0)}$ and $\alpha^3 \geq (32\rho^2 z \sqrt{2\Phi(\boldsymbol{\theta}^0)})/\eta_O$, then with probability at least $1 - 8e^{-d_m/2} - \delta - \exp(-\Omega((N-1)^{-\hat{c}}d_s^{-1}))$ for proper $\delta$, GD converges to a global minimum as follows for a sufficiently small step size $\gamma \leq 1/k$ with $k$ as a constant depending on $(\xi_Q, \xi_K, \xi_V, \eta_O, \Phi(\boldsymbol{\theta}^0), \rho, d_m^{-1/2})$:

$$\Phi(\boldsymbol{\theta}^t) \leq \left(1 - \gamma \frac{\alpha^2}{2}\right)^t \Phi(\boldsymbol{\theta}^0), \forall t \geq 0$$

where $M = \max(\xi_V R_O^{-1}, \eta_O R_V^{-1}, (2/\sqrt{d_m}) R_X^2 d_s \xi_K \xi_V \eta_O R_Q^{-1}, (2/\sqrt{d_m}) R_X^2 d_s \xi_Q \xi_V \eta_O R_K^{-1})$ and $\rho \triangleq N^{1/2} d_s^{3/2} R_X$, $z \triangleq \eta_O^2 (1 + (4/d_m) R_X^4 d_s^2 \xi_V^2 (\xi_Q^2 + \xi_K^2))$.

We can observe that $\Phi(\boldsymbol{\theta}^t)$ decays exponentially as training proceeds. This implies the following bound;

$$\Phi(\boldsymbol{\theta}^t) \leq \Phi(\boldsymbol{\theta}^0), \quad \forall t \geq 0$$

From the first condition i.e., $\alpha^2 \geq 8\rho M \sqrt{2\Phi(\boldsymbol{\theta}^0)}$, we can say that $\Phi(\boldsymbol{\theta}^0) \leq \alpha^4/(128\rho^2 M^2)$. We therefore end up with the bound below;

$$\Phi(\boldsymbol{\theta}^t) \leq \frac{\alpha^4}{128\rho^2 M^2}, \quad \forall t \geq 0$$

From the second condition i.e., $\alpha^3 \geq (32\rho^2 z \sqrt{2\Phi(\boldsymbol{\theta}^0)})/\eta_O$, we can say that $\Phi(\boldsymbol{\theta}^0) \leq (\alpha^6 \eta_O^2)/(2048\rho^4 z^2)$. We therefore end up with the bound below;

$$\Phi(\boldsymbol{\theta}^t) \leq \frac{\alpha^6 \eta_O^2}{2048\rho^4 z^2}, \quad \forall t \geq 0$$

Combining the two bounds on $\Phi(\boldsymbol{\theta}^t)$, we obtain the final bound as;

$$\Phi(\boldsymbol{\theta}^t) \leq \min\left(\frac{\alpha^4}{128\rho^2 M^2}, \frac{\alpha^6 \eta_O^2}{2048\rho^4 z^2}\right), \quad \forall t \geq 0$$

Our empirical loss i.e., $\mathcal{L}_S(f_{\boldsymbol{\theta}^t}) = \frac{1}{N}\sum_{n=1}^{N} \ell(f_{\boldsymbol{\theta}^t}(\boldsymbol{X}_n), y_n)$ for all $t > 0$ can be bounded as follows;

$$\mathcal{L}_S(f_{\boldsymbol{\theta}^t}) \leq \frac{1}{N}\sum_{n=1}^{N}\left(\ell(f_{\boldsymbol{\theta}^t}(\boldsymbol{X}_n), y_n) - \ell(y_n, y_n)\right)$$

$$\leq \frac{1}{N}\sum_{n=1}^{N}|f_{\boldsymbol{\theta}^t}(\boldsymbol{X}_n) - y_n| \qquad \text{because } \ell(\cdot, \cdot) \text{ is 1-Lipschitz in the first argument}$$

$$\leq \frac{1}{\sqrt{N}}\|\boldsymbol{f}_{\boldsymbol{\theta}^t} - \boldsymbol{y}\|_2$$

$$= \sqrt{\frac{2\Phi(\boldsymbol{\theta}^t)}{N}}$$

$$\leq \min\left(\frac{\alpha^2}{8\rho M\sqrt{N}}, \frac{\alpha^3 \eta_O}{32\rho^2 z\sqrt{N}}\right)$$

where $M = \max(\xi_V R_O^{-1}, \eta_O R_V^{-1}, (2/\sqrt{d_m}) R_X^2 d_s \xi_K \xi_V \eta_O R_Q^{-1}, (2/\sqrt{d_m}) R_X^2 d_s \xi_Q \xi_V \eta_O R_K^{-1})$ and $\rho \triangleq N^{1/2} d_s^{3/2} R_X$, $z \triangleq \eta_O^2 (1 + (4/d_m) R_X^4 d_s^2 \xi_V^2 (\xi_Q^2 + \xi_K^2))$.

Upper bounding $\xi_O, \xi_V, \xi_K, \xi_Q$ using $\eta_O, \eta_V, \eta_K, \eta_Q$, the upper bound on the empirical loss for all training steps can therefore be written as;

$$\mathcal{L}_S(f_{\boldsymbol{\theta}^t}) \leq \min\left(\frac{\alpha^2}{8\rho \hat{M} \sqrt{N}}, \frac{\alpha^3 \eta_O}{32\rho^2 \hat{z} \sqrt{N}}\right)$$

where $\hat{M} = \max(\eta_V R_O^{-1}, \eta_O R_V^{-1}, (2/\sqrt{d_m}) R_X^2 d_s \eta_K \eta_V \eta_O R_Q^{-1}, (2/\sqrt{d_m}) R_X^2 d_s \eta_Q \eta_V \eta_O R_K^{-1})$ and $\hat{z} \triangleq \eta_O^2 (1 + (4/d_m) R_X^4 d_s^2 \eta_V^2 (\eta_Q^2 + \eta_K^2))$. This completes the proof. $\square$

## 4.4 Main result

With both the Rademacher complexity and empirical loss bounds in place, we now combine these results to obtain our main theorem i.e., a high-probability upper bound on the true loss (expected generalization error) of shallow Transformer models under bounded parameter drift. The theorem demonstrates that the true loss decreases with the number of training samples $N$ and depends explicitly on the initialization scale, data structure, and model dimension $d_m$. The bound captures the essence of lazy training: when model parameters remain near initialization, generalization behavior aligns with that of linearized models governed by the neural tangent kernel (NTK). While this result establishes a rigorous theoretical foundation for shallow Transformers, it also highlights the need for further analysis to extend such guarantees to deeper and more expressive models.

**Theorem 5.** *Suppose that we have $\eta_V = \|\boldsymbol{W}_V^0\|_F + R_V, \eta_O = \|\boldsymbol{w}_O^0\|_2 + R_O, \eta_K = \|\boldsymbol{W}_K^0\|_F + R_K, \eta_Q = \|\boldsymbol{W}_Q^0\|_F + R_Q, \xi_Q = \|\boldsymbol{W}_Q^0\|_2 + R_Q, \xi_K = \|\boldsymbol{W}_K^0\|_2 + R_K, \xi_V = \|\boldsymbol{W}_V^0\|_2 + R_V$ where $R_O, R_V, R_K, R_Q$ remain as defined earlier. Under assumption 3, if $d_m \geq \widetilde{\Omega}(N^3)$, $\alpha^2 \geq 8\rho M \sqrt{2\Phi(\boldsymbol{\theta}^0)}, \alpha^3 \geq (32\rho^2 z \sqrt{2\Phi(\boldsymbol{\theta}^0)})/\eta_O$ and $\ell(\boldsymbol{\theta})$ is any loss function which is 1-lipschitz in the first argument, then with probability at least $1 - 8e^{-d_m/2} - 2\delta - \exp(-\Omega((N-1)^{-\hat{c}} d_s^{-1}))$, if the transformer model is trained using Gradient Descent with small step size $\gamma \leq 1/k$ where $k$ is a constant depending on $(\xi_Q, \xi_K, \xi_V, \eta_O, \ell(\boldsymbol{\theta}^0), \rho, d_m^{-1/2})$, the true loss $L_{\mathcal{D}}(f)$ can be bounded as follows;*

$$L_{\mathcal{D}}(f) \lesssim \min\left(\frac{\alpha^2}{8\rho \hat{M} \sqrt{N}}, \frac{\alpha^3 \eta_O}{32\rho^2 \hat{z} \sqrt{N}}\right) + \mathcal{O}\left(\sqrt{\frac{P}{N}}\left(1 + \log\left(A\sqrt{\frac{N}{P}}\right)\right) + \sqrt{\frac{\log\frac{R}{\delta}}{N}}\right)$$

*where $\widetilde{\Omega}$ omits the logarithmic factor, $\lesssim$ hides logarithmic dependencies on quantities besides $N$, $d_s$ and $\delta$ and the other quantities are defined as follows; $\rho \triangleq N^{1/2} d_s^{3/2} R_X$,*
*$z \triangleq \eta_O^2 (1 + (4/d_m) R_X^4 d_s^2 \xi_V^2 (\xi_Q^2 + \xi_K^2)), \hat{z} \triangleq \eta_O^2 (1 + (4/d_m) R_X^4 d_s^2 \eta_V^2 (\eta_Q^2 + \eta_K^2)),$*
*$M = \max(\xi_V R_O^{-1}, \eta_O R_V^{-1}, (2/\sqrt{d_m}) R_X^2 d_s \xi_K \xi_V \eta_O R_Q^{-1}, (2/\sqrt{d_m}) R_X^2 d_s \xi_Q \xi_V \eta_O R_K^{-1}),$*
*$\hat{M} = \max(\eta_V R_O^{-1}, \eta_O R_V^{-1}, (2/\sqrt{d_m}) R_X^2 d_s \eta_K \eta_V \eta_O R_Q^{-1}, (2/\sqrt{d_m}) R_X^2 d_s \eta_Q \eta_V \eta_O R_K^{-1}),$*
*$A = \eta_O \eta_V (\sqrt{d_s} R_X)$ and $P = (\sqrt{d_s} R_X)^2 \left(\left(\sqrt{d_m} \eta_V\right)^{\frac{2}{3}} + \left(\sqrt{d_m} \eta_K \eta_Q \eta_V\right)^{\frac{2}{3}}\right)^3 \log(N d_s).$*

*Proof.* Recall that we defined our hypothesis class as follows;

$$\mathcal{F}_R^{\boldsymbol{\theta}^0} = \left\{f_{\boldsymbol{\theta}}(\boldsymbol{X}_n) : \|\boldsymbol{\theta}^{t+1} - \boldsymbol{\theta}^0\|_2 \leq R\right\}$$

Let us set $R_i = i$ for $i \in \{1, 2, \ldots, R\}$. This means that we can define a class of models whose parameter norm is bounded as $\|\boldsymbol{\theta}^{t+1} - \boldsymbol{\theta}^0\|_2 \leq R_i$ for $i \in \{1, 2, \ldots, R\}$ as follows;

$$\mathcal{F}_{R_i}^{\boldsymbol{\theta}^0} = \left\{f_{\boldsymbol{\theta}}(\boldsymbol{X}_n) : \|\boldsymbol{\theta}^{t+1} - \boldsymbol{\theta}^0\|_2 \leq R_i\right\}$$

From Rademacher complexity and a union bound over a finite set of $R_i$'s, for any random initialization $(\boldsymbol{\theta}^0)$, with probability at least $1 - \delta$ over the sample $S = \{(\boldsymbol{X}_n, y_n)\}_{n=1}^N$ of size $N$, we have that;

$$\sup_{f \in \mathcal{F}_{R_i}^{\boldsymbol{\theta}^0}} \{L_{\mathcal{D}}(f) - L_S(f)\} \leq 2\mathcal{R}_S(\mathcal{F}_{R_i}^{\boldsymbol{\theta}^0}) + \sqrt{\frac{\log \frac{2R}{\delta}}{2N}}$$

for all $i \in \{1, 2, 3, \ldots, R\}$. Note that $R_i \leq R$ for all $i \in \{1, 2, \ldots, R\}$ which implies that $\mathcal{R}_S(\mathcal{F}_{R_i}^{\boldsymbol{\theta}^0}) \leq \mathcal{R}_S(\mathcal{F}_R^{\boldsymbol{\theta}^0})$ for any $i \in \{1, 2, \ldots, R\}$. This gives us the following bound on $\mathcal{R}_S(\mathcal{F}_{R_i}^{\boldsymbol{\theta}^0})$ for all $i \in \{1, 2, \ldots, R\}$;

$$\mathcal{R}_S(\mathcal{F}_{R_i}^{\boldsymbol{\theta}^0}) \lesssim \mathcal{O}\left(\sqrt{\frac{P}{N}}\left(1 + \log\left(A\sqrt{\frac{N}{P}}\right)\right)\right)$$

where $P = (\sqrt{d_s}R_X)^2 \left(\left(\sqrt{d_m}\eta_V\right)^{\frac{2}{3}} + \left(\sqrt{d_m}\eta_K\eta_Q\eta_V\right)^{\frac{2}{3}}\right)^3 \log(Nd_s)$ and $A = \eta_O\eta_V(\sqrt{d_s}R_X)$. From lemma 4, with probability at least $1 - 8e^{-d_m/2} - \delta - \exp(-\Omega((N-1)^{-\hat{c}}d_s^{-1}))$, the training loss for our transformer model can be bounded as follows for all $t > 0$;

$$\mathcal{L}_S(f_{\boldsymbol{\theta}^t}) \leq \min\left(\frac{\alpha^2}{8\rho\hat{M}\sqrt{N}}, \frac{\alpha^3\eta_O}{32\rho^2\hat{z}\sqrt{N}}\right)$$

where $\rho \triangleq N^{1/2}d_s^{3/2}R_X$, $\hat{z} \triangleq \eta_O^2(1 + (4/d_m)R_X^4d_s^2\eta_V^2(\eta_Q^2 + \eta_K^2))$ and
$\hat{M} = \max(\eta_V R_O^{-1}, \eta_O R_V^{-1}, (2/\sqrt{d_m})R_X^2 d_s\eta_K\eta_V\eta_O R_Q^{-1}, (2/\sqrt{d_m})R_X^2 d_s\eta_Q\eta_V\eta_O R_K^{-1})$.
Putting everything together, with probability atleast $1 - 8e^{-d_m/2} - 2\delta - \exp(-\Omega((N-1)^{-\hat{c}}d_s^{-1}))$, we have that;

$$L_{\mathcal{D}}(f) \lesssim \min\left(\frac{\alpha^2}{8\rho\hat{M}\sqrt{N}}, \frac{\alpha^3\eta_O}{32\rho^2\hat{z}\sqrt{N}}\right) + \mathcal{O}\left(\sqrt{\frac{P}{N}}\left(1 + \log\left(A\sqrt{\frac{N}{P}}\right)\right) + \sqrt{\frac{\log \frac{R}{\delta}}{N}}\right)$$

where $\lesssim$ hides logarithmic dependencies on quantities besides $N$, $d_s$ and $\delta$. This completes the proof. $\qquad\square$

## 4.5 Discussion

Our main theorem provides a generalization bound for a class of Transformer models whose weights remain close to their initialization during training. This bounded-drift assumption effectively constrains the training dynamics to what is commonly referred to as the *lazy training regime*. In this setting, the model behaves similarly to its linearized form around initialization, which has been extensively studied in the context of neural tangent kernels (NTK).

The implication of this assumption is that the results derived here apply most accurately to wide and shallow Transformers trained with sufficiently small learning rates that prevent the parameters from deviating significantly from their initial values. Such a setting captures the early or near-linear training phase of overparameterized models, where the NTK remains nearly constant and generalization can be controlled through classical complexity measures such as the Rademacher complexity.

Compared to prior results on generalization bounds for neural networks under the lazy regime (e.g., for two-layer networks and linearized models), our bound maintains a similar dependence on the sample size $N$ and model width $d_m$. Specifically, the $\mathcal{O}\left(\sqrt{\frac{P}{N}}\log(A\sqrt{\frac{N}{P}})\right)$ term scales analogously to existing NTK-based results, while the dependence on $\alpha$ and $\rho$ in the empirical loss component reflects the conditioning of the pre-activation features and the interaction between model width and sequence length. However, unlike the more general results that extend to deep networks or non-lazy regimes through complex stability analyses or overparameterization assumptions, our result explicitly applies to *single-layer* (shallow) Transformer architectures only.

We also note that relaxing the bounded-drift assumption to capture non-lazy or feature-learning regimes remains an open problem. In such regimes, weights undergo significant changes during training, resulting in evolving representations and coupling across layers. Extending our analysis to this setting would require novel techniques to handle the dynamic evolution of the NTK or an equivalent representation matrix. Similarly, extending the bound to deeper multi-layer architectures would require careful control of layerwise dependencies, possibly through hierarchical complexity bounds or layerwise Lipschitz control arguments.

Overall, our results contribute to the growing theoretical understanding of Transformers under simplified but analytically tractable conditions. They highlight the relationship between network width, data complexity, and generalization under bounded parameter drift, reinforcing the intuition that in the lazy regime, wide and shallow Transformers behave as near-linear models governed by their initialization structure.

**Comparison with other norm-based Transformer bounds.** Our result differs substantively from norm-based bounds tailored to Transformers. Edelman et al. (2021) obtain a gap bound that scales only *logarithmically* with sequence length by analyzing bounded-norm self-attention and the sparse variable creation inductive bias, while Trauger & Tewari (2024) strengthen this line by proving *sequence-length-independent* gap bounds via a covering-number analysis that upper-bounds the Rademacher complexity of Transformer classes and applies also to masked-token training objectives. In contrast, our theorem provides a *true-loss* bound that explicitly couples a data-dependent optimization guarantee (lazy-regime convergence for a shallow Transformer) with a capacity term. Practically, this means (i) norm-based results (Edelman et al., 2021; Trauger & Tewari, 2024) can be evaluated after training completes by plugging in norms to bound $|\mathcal{L}_{\mathcal{D}}(f) - \mathcal{L}_S(f)|$, and they cleanly characterize how sequence length enters (logarithmically or not at all), whereas (ii) our bound ties generalization to optimization-side quantities such as $\alpha$, $\rho$, and $(\eta_Q, \eta_K, \eta_V, \eta_O)$ under bounded drift and thus characterizes when low *true* risk is guaranteed without first computing $\mathcal{L}_S(f)$. Conceptually, the results are complementary: norm-based bounds offer architecture-wide, training-regime-agnostic control of the generalization gap (with refined sequence-length dependence), while our analysis isolates the lazy, shallow regime and provides optimization-aware control of the *level* of the true loss.

**Relation to PAC-Bayes and stability-based bounds.** Compared to PAC-Bayes or compression bounds for LLMs (Lotfi et al., 2023; Zhou et al., 2018), our result targets a different regime and objective: we operate under a bounded-drift (lazy) assumption and obtain a *true-loss* bound for a *single-layer* Transformer trained by gradient descent, with explicit dependence on quantities like $\alpha$, $\rho$, and $(\eta_Q, \eta_K, \eta_V, \eta_O)$. PAC-Bayes bounds, in contrast, typically control $|\mathcal{L}_{\mathcal{D}}(f) - \mathcal{L}_S(f)|$ via a data-informed posterior and compression, and have been instantiated for large, deep, pretrained language models without requiring lazy dynamics. Stability-based results for Transformers (e.g., in-context learning) quantify sensitivity to sample perturbations through stability coefficients (Li et al., 2023b), again bounding the generalization gap rather than the true loss. Thus, our contribution is complementary: it connects *optimization-driven* lazy training (bounded drift + shallow width assumptions) to generalization, while PAC-Bayes and stability provide depth-agnostic, training-regime-agnostic controls on the gap. Extending our approach to remove the lazy assumption or to handle deeper stacks could help bridge these viewpoints, potentially yielding hybrid bounds that combine optimization-aware true-loss control with posterior- or stability-based gap terms (Yao et al., 2025).

## 5   Limitations

While our analysis provides a rigorous generalization bound for shallow Transformer models trained under bounded parameter drift, it is important to clarify the scope and limitations of the results.

**Loose nature of Rademacher-based bounds.** Our generalization analysis relies on upper bounds on the empirical Rademacher complexity of a constrained hypothesis class. It is well known that classical Rademacher complexity bounds for neural networks are often quantitatively loose, especially for large, overparameterized models such as Transformers. In particular, these bounds are typically worst-case and may not tightly reflect the effective capacity of models encountered in practice. Consequently, while our results establish a principled scaling behavior with respect to sample size, width, and norm constraints, the re-

sulting bounds should be interpreted as qualitative guarantees rather than sharp predictors of empirical performance.

**Restrictive nature of the lazy (bounded-drift) regime.** A central assumption in our analysis is that the Transformer parameters remain close to their initialization throughout training, corresponding to the lazy or NTK regime. Although this assumption enables tractable theoretical analysis and connects our results to classical kernel-based generalization theory, it does not fully capture the training dynamics observed in practical Transformer training, where substantial feature learning and representation drift typically occur. As a result, our guarantees are most relevant for wide, shallow Transformers trained with sufficiently small learning rates, or for early training phases before significant departure from initialization.

**Comparison with existing analyses of Transformer training dynamics.** There exists a growing body of prior work that studies Transformer training dynamics using alternative analytical tools, including mean-field limits, dynamical systems analyses, signal propagation, and task-specific convergence characterizations. These approaches often provide finer-grained insights into representation learning, optimization trajectories, or task-dependent inductive biases beyond what is captured by Rademacher-based complexity measures. Our work is not intended to subsume these analyses; rather, it complements them by offering a unified generalization bound that explicitly couples optimization guarantees (via lazy-regime convergence) with statistical capacity control for a well-defined hypothesis class.

**Architectural and depth limitations.** Our results apply only to *single-layer* (shallow) Transformer architectures. Extending the analysis to deeper stacks introduces significant technical challenges, including the control of layerwise interactions, accumulation of approximation errors, and evolving attention representations. Addressing these challenges would likely require new techniques beyond the bounded-drift framework employed here, such as hierarchical complexity bounds, stability arguments, or depth-dependent norm control.

**Outlook.** Despite these limitations, we view our analysis as a useful step toward understanding how optimization dynamics and generalization interact in Transformer models under analytically tractable conditions. An important direction for future work is to relax the bounded-drift assumption and to integrate alternative toolssuch as stability, PAC-Bayes, or feature-learning analyses with optimization-aware bounds, in order to more faithfully capture the behavior of modern deep Transformers trained beyond the lazy regime.

## 6 Conclusion

In summary, we established a generalization bound for shallow Transformer models trained in the bounded-drift (lazy) regime, where the model parameters remain close to their initialization throughout training. By combining Rademacher complexity analysis with an upper bound on the empirical loss, we obtained a probabilistic bound on the true loss that decreases with the number of samples $N$ and depends explicitly on model and data parameters.

Our theoretical results align with existing findings for wide, overparameterized models analyzed under the NTK framework, but they are specific to single-layer Transformers. This limitation ensures that our claims remain within the theoretical scope supported by the bounded-drift assumption. Extending the analysis to deeper architectures or non-lazy training regimes (where substantial feature learning occurs) remains an important direction for future research.

Through this work, we provide a rigorous foundation for understanding how bounded-drift dynamics influence generalization in Transformer models and set the stage for future extensions that aim to capture the richer behavior of modern, deeper architectures trained beyond the lazy regime.

**Acknowledgments**

Anirban Dasgupta acknowledges the support by SERB MATRICS and SERB CRG grants and the support from N Rama Rao Chair Professorship.

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

# A  Proof of lemma 2

We want to obtain an upper bound on $\mathcal{R}_S(\mathcal{G}_R^{\boldsymbol{\theta}^0})$ where $\mathcal{G}_R^{\boldsymbol{\theta}^0}$ is defined as follows;

$$\mathcal{G}_R^{\boldsymbol{\theta}^0} := \left\{ (\boldsymbol{X}^{(i,:)})^\top \longrightarrow \boldsymbol{w}_O^\top \sigma_r \left( \boldsymbol{W}_V \boldsymbol{X}_n^\top \sigma_s \left( \frac{\boldsymbol{X}_n \boldsymbol{W}_K^\top \boldsymbol{W}_Q (\boldsymbol{X}_n^{(i,:)})^\top}{\sqrt{d_m}} \right) \right) : \begin{array}{c} \|\boldsymbol{w}_O\|_2 \le \eta_O \\ \|\boldsymbol{W}_V\|_F \le \eta_V \\ \left\| \frac{\boldsymbol{w}_K^\top \boldsymbol{w}_Q}{\sqrt{d_m}} \right\|_F \le \frac{\eta_K \eta_Q}{\sqrt{d_m}} \end{array} \right\}$$

Let's begin by defining the following bounds on the matrices;

$$\|\boldsymbol{W}_V\|_2 \le \|\boldsymbol{W}_V^0\|_2 + b_V \le \eta_V$$

$$\|\boldsymbol{W}_V\|_{2,1} \le \|\boldsymbol{W}_V^0\|_{2,1} + B_V \le \sqrt{d_m}\eta_V$$

$$\left\| \frac{\boldsymbol{W}_K^\top \boldsymbol{W}_Q}{\sqrt{d_m}} \right\|_{2,1} \le \frac{\left( \|(\boldsymbol{W}_K^0)^\top\|_{1,2} + B_K \right) \left( \|\boldsymbol{W}_Q^0\|_{2,1} + B_Q \right)}{\sqrt{d_m}} \le \frac{d_m \eta_K \eta_Q}{\sqrt{d_m}} = \sqrt{d_m}\eta_K \eta_Q$$

$$\|\boldsymbol{X}_n^\top\|_{2,\infty} \le B_X \le \|\boldsymbol{X}_n\|_F \le \sqrt{d_s} R_X \quad \forall n \in [N]$$

where $b_V, B_V, B_K, B_Q, B_X$ are some positive constants and $R_O, R_V, R_K, R_Q, R_X$ remain as defined earlier. The norm $\|\cdot\|_{2,1}$ interpreted as first taking the $\ell_2$-norm for each column of a matrix and then summing these column norms.

Define another class $\mathcal{G}_B^{\boldsymbol{\theta}^0}$ as shown below;

$$\mathcal{G}_B^{\boldsymbol{\theta}^0} := \left\{ (\boldsymbol{X}^{(i,:)})^\top \longrightarrow \boldsymbol{w}_O^\top \sigma_r \left( \boldsymbol{W}_V \boldsymbol{X}_n^\top \sigma_s \left( \boldsymbol{X}_n \boldsymbol{W}_K^\top \boldsymbol{W}_Q (\boldsymbol{X}_n^{(i,:)})^\top \right) \right) : \begin{array}{c} \|\boldsymbol{w}_O\|_2 \le \eta_O \\ \|\boldsymbol{W}_V\|_2 \le \|\boldsymbol{W}_V^0\|_2 + b_V \\ \|\boldsymbol{W}_V\|_{2,1} \le \|\boldsymbol{W}_V^0\|_{2,1} + B_V \\ \left\| \frac{\boldsymbol{w}_K^\top \boldsymbol{w}_Q}{\sqrt{d_m}} \right\|_{2,1} \le \frac{\left( \|(\boldsymbol{W}_K^0)^\top\|_{1,2} + B_K \right) \left( \|\boldsymbol{W}_Q^0\|_{2,1} + B_Q \right)}{\sqrt{d_m}} \end{array} \right\}$$

The following lemma gives an upper bound on the log covering number of the class $\mathcal{G}_B^{\boldsymbol{\theta}^0}$;

**Lemma 6.** ((Edelman et al., 2021) Corollary 4.5). *For any fixed $\epsilon > 0$ and $\boldsymbol{X}_1, \ldots, \boldsymbol{X}_N \in \mathbb{R}^{d_s \times d}$ such that $\|\boldsymbol{X}_n^\top\|_{2,\infty} \le B_X$ for all $n \in [N]$, the covering number of $\mathcal{G}_B^{\boldsymbol{\theta}^0}$ satisfies the bound given below;*

$$\log \mathcal{N}_\infty(\mathcal{G}_B^{\boldsymbol{\theta}^0}; \epsilon; \{\boldsymbol{X}_n\}_{n=1}^N, \|\cdot\|_2)$$

$$\lesssim B_X^2 \cdot \frac{\left( \left( \|\boldsymbol{W}_V^0\|_{2,1} + B_V \right)^{\frac{2}{3}} + \left( \left( \frac{\left( \|(\boldsymbol{W}_K^0)^\top\|_{1,2} + B_K \right) \left( \|\boldsymbol{W}_Q^0\|_{2,1} + B_Q \right)}{\sqrt{d_m}} \right) \left( \|\boldsymbol{W}_V^0\|_2 + b_V \right) \right)^{\frac{2}{3}} \right)^3}{\epsilon^2} \cdot \log(N d_s)$$

*where $\lesssim$ hides logarithmic dependencies on quantities besides $N$ and $d_s$.*

Upper bounding the norms $\|\cdot\|_{2,1}$ and $\|\cdot\|_{2,\infty}$ using the Frobenius norm, $\|\cdot\|_F$, we end up with;

$$\log \mathcal{N}_\infty(\mathcal{G}_R^{\boldsymbol{\theta}^0}; \epsilon; \{\boldsymbol{X}_n\}_{n=1}^N, \|\cdot\|_2) \lesssim (\sqrt{d_s} R_X)^2 \cdot \frac{\left( \left( \sqrt{d_m}\eta_V \right)^{\frac{2}{3}} + \left( \sqrt{d_m}\eta_K \eta_Q \eta_V \right)^{\frac{2}{3}} \right)^3}{\epsilon^2} \cdot \log(N d_s)$$

This can also be written as;

$$\log \mathcal{N}_\infty(\mathcal{G}_R^{\boldsymbol{\theta}^0}; \epsilon; \{\boldsymbol{X}_n\}_{n=1}^N, \|\cdot\|_2) \lesssim \frac{P}{\epsilon^2}$$

where $P = (\sqrt{d_s} R_X)^2 \left( \left( \sqrt{d_m}\eta_V \right)^{\frac{2}{3}} + \left( \sqrt{d_m}\eta_K \eta_Q \eta_V \right)^{\frac{2}{3}} \right)^3 \log(N d_s)$.

We can now write the bound on the Rademacher complexity $\mathcal{R}_S(\mathcal{G}_R^{\boldsymbol{\theta}^0})$ as follows for some constant $c > 0$ and $|f| \leq A$ for all $f \in \mathcal{G}_R^{\boldsymbol{\theta}^0}$;

$$
\mathcal{R}_S(\mathcal{G}_R^{\boldsymbol{\theta}^0}) \leq c \cdot \inf_{\delta \geq 0} \left( \delta + \int_\delta^A \sqrt{\frac{\log \mathcal{N}_\infty(\mathcal{G}_R^{\boldsymbol{\theta}^0}; \epsilon; \{\boldsymbol{X}_n\}_{n=1}^N; \|\cdot\|_2)}{N}} d\epsilon \right)
$$

$$
\lesssim c \cdot \inf_{\delta \geq 0} \left( \delta + \int_\delta^A \sqrt{\frac{P}{\epsilon^2 N}} d\epsilon \right)
$$

$$
= c \cdot \inf_{\delta \geq 0} \left( \delta + \sqrt{\frac{P}{N}} \int_\delta^A \frac{1}{\epsilon} d\epsilon \right)
$$

$$
= c \cdot \inf_{\delta \geq 0} \left( \delta + \sqrt{\frac{P}{N}} \log\left(\frac{A}{\delta}\right) \right)
$$

$$
= c\sqrt{\frac{P}{N}} \left( 1 + \log\left( A\sqrt{\frac{N}{P}} \right) \right)
$$

Note that $|f| \leq A$ for all $f \in \mathcal{G}_R^{\boldsymbol{\theta}^0}$. $A$ can be obtained as follows;

$$
\left| \boldsymbol{w}_O^\top \sigma_r \left( \boldsymbol{W}_V \boldsymbol{X}_n^\top \sigma_s \left( \frac{\boldsymbol{X}_n \boldsymbol{W}_K^\top \boldsymbol{W}_Q (\boldsymbol{X}_n^{(i,:)})^\top}{\sqrt{d_m}} \right) \right) \right|
$$

$$
\leq \|\boldsymbol{w}_O\|_2 \left\| \sigma_r \left( \boldsymbol{W}_V \boldsymbol{X}_n^\top \sigma_s \left( \frac{\boldsymbol{X}_n \boldsymbol{W}_K^\top \boldsymbol{W}_Q (\boldsymbol{X}_n^{(i,:)})^\top}{\sqrt{d_m}} \right) \right) \right\|_2
$$

$$
= \|\boldsymbol{w}_O\|_2 \left\| \sigma_r \left( \boldsymbol{W}_V \boldsymbol{X}_n^\top \sigma_s \left( \frac{\boldsymbol{X}_n \boldsymbol{W}_K^\top \boldsymbol{W}_Q (\boldsymbol{X}_n^{(i,:)})^\top}{\sqrt{d_m}} \right) \right) \right\|_2
$$

$$
\leq \|\boldsymbol{w}_O\|_2 \left\| \boldsymbol{W}_V \boldsymbol{X}_n^\top \sigma_s \left( \frac{\boldsymbol{X}_n \boldsymbol{W}_K^\top \boldsymbol{W}_Q (\boldsymbol{X}_n^{(i,:)})^\top}{\sqrt{d_m}} \right) \right\|_2 \quad (\text{because } \|\sigma_r(\boldsymbol{z})\|_2 \leq \|\boldsymbol{z}\|_2)
$$

$$
\leq \|\boldsymbol{w}_O\|_2 \|\boldsymbol{W}_V\|_2 \left\| \boldsymbol{X}_n^\top \sigma_s \left( \frac{\boldsymbol{X}_n \boldsymbol{W}_K^\top \boldsymbol{W}_Q (\boldsymbol{X}_n^{(i,:)})^\top}{\sqrt{d_m}} \right) \right\|_2
$$

$$
\leq \|\boldsymbol{w}_O\|_2 \|\boldsymbol{W}_V\|_2 \|\boldsymbol{X}_n\|_2 \left\| \sigma_s \left( \frac{\boldsymbol{X}_n \boldsymbol{W}_K^\top \boldsymbol{W}_Q (\boldsymbol{X}_n^{(i,:)})^\top}{\sqrt{d_m}} \right) \right\|_2
$$

$$
\leq \|\boldsymbol{w}_O\|_2 \|\boldsymbol{W}_V\|_2 \|\boldsymbol{X}_n\|_2 \quad (\text{because } \|\sigma_s(\boldsymbol{z})\|_2 \leq \|\sigma_s(\boldsymbol{z})\|_1 = 1)
$$

$$
\leq \|\boldsymbol{w}_O\|_2 \|\boldsymbol{W}_V\|_F \|\boldsymbol{X}_n\|_F
$$

$$
\leq (\|\boldsymbol{w}_O^0\|_2 + R_O)(\|\boldsymbol{W}_V^0\|_F + R_V)(\sqrt{d_s} R_X)
$$

$$
= \eta_O \eta_V (\sqrt{d_s} R_X)
$$

This means that $A = \eta_O \eta_V (\sqrt{d_s} R_X)$. $\qquad\qquad\qquad\qquad\qquad\qquad\qquad \square$

## B    Experiments

### B.1    Image Classification

We use the transformer model defined in section 3.1.2 to perform classification of images. From MNIST dataset, we extract the images belonging to classes 0 and 1 and create our new dataset. Each image of size $28 \times 28$ is broken into tokens each of dimension $d = 64$. The main goal of the experiments is to demonstrate that the test loss of the trained transformer model decreases with increasing number of of samples i.e., $N = 400, N = 1200$ and $N = 10000$. This trend holds for all the values of model dimension which we tested

i.e., $d_m = 64, d_m = 1024$ and $d_m = 4096$. The learning rate used is 0.1, the optimization algorithm is batch gradient descent and the loss function is the cross-entropy loss. The results for the experiments are presented below. Each figure 1-3 shows the training loss and test loss of the transformer model as training proceeds.

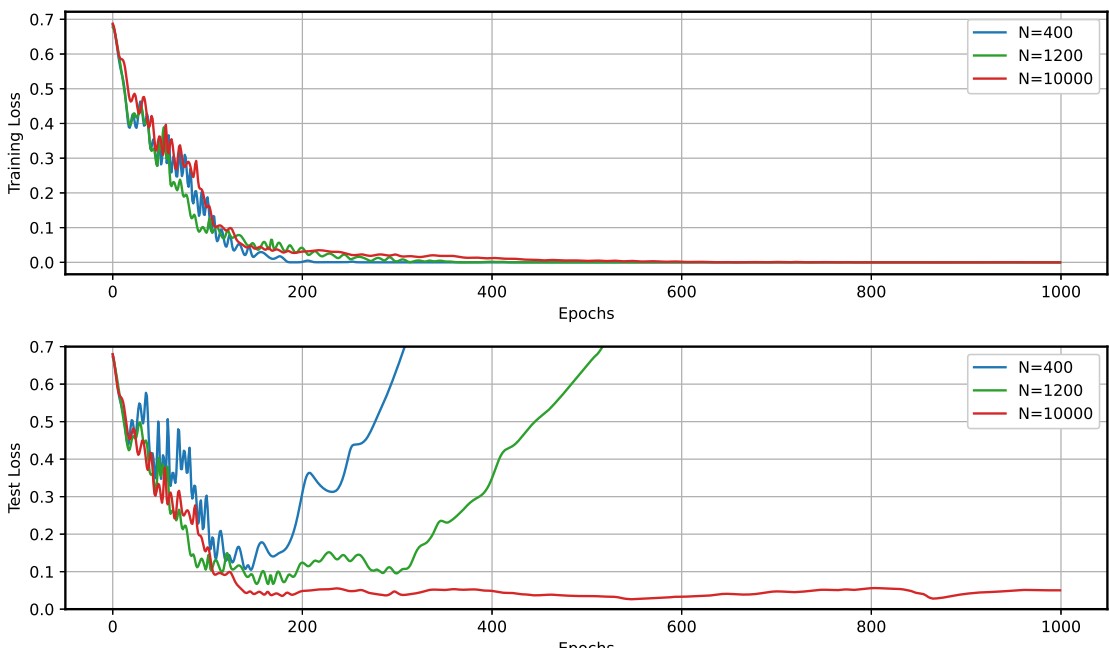

Figure 1: Evolution of training loss (top) and test loss(bottom) for each epoch of training for model dimension $d_m = 64$.

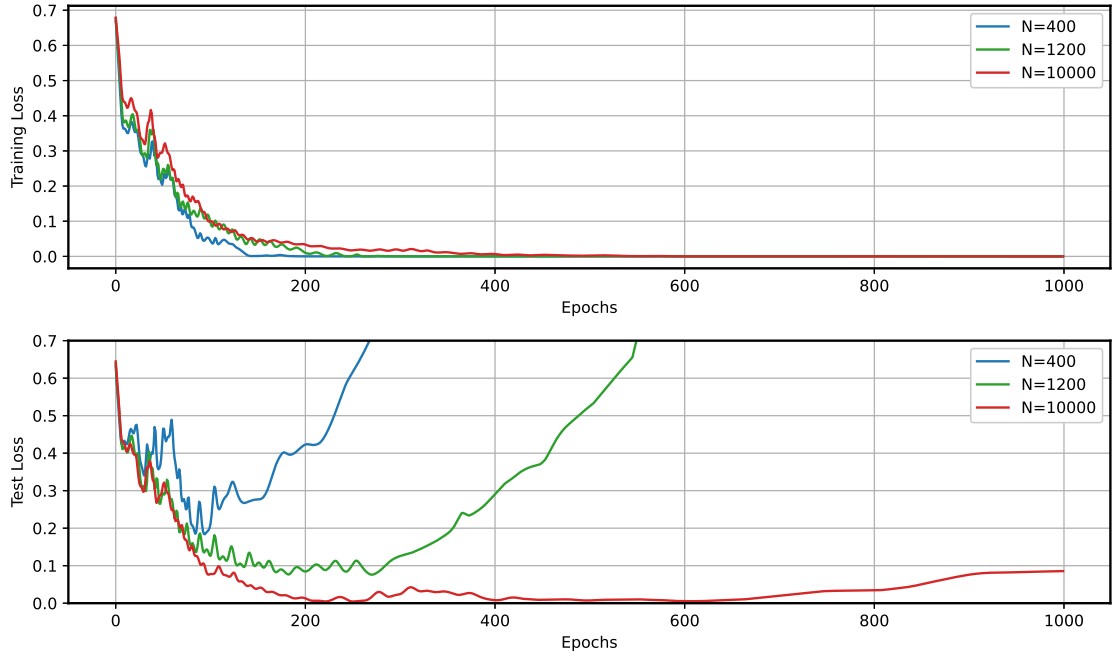

Figure 2: Evolution of training loss (top) and test loss(bottom) for each epoch of training for model dimension $d_m = 1024$.

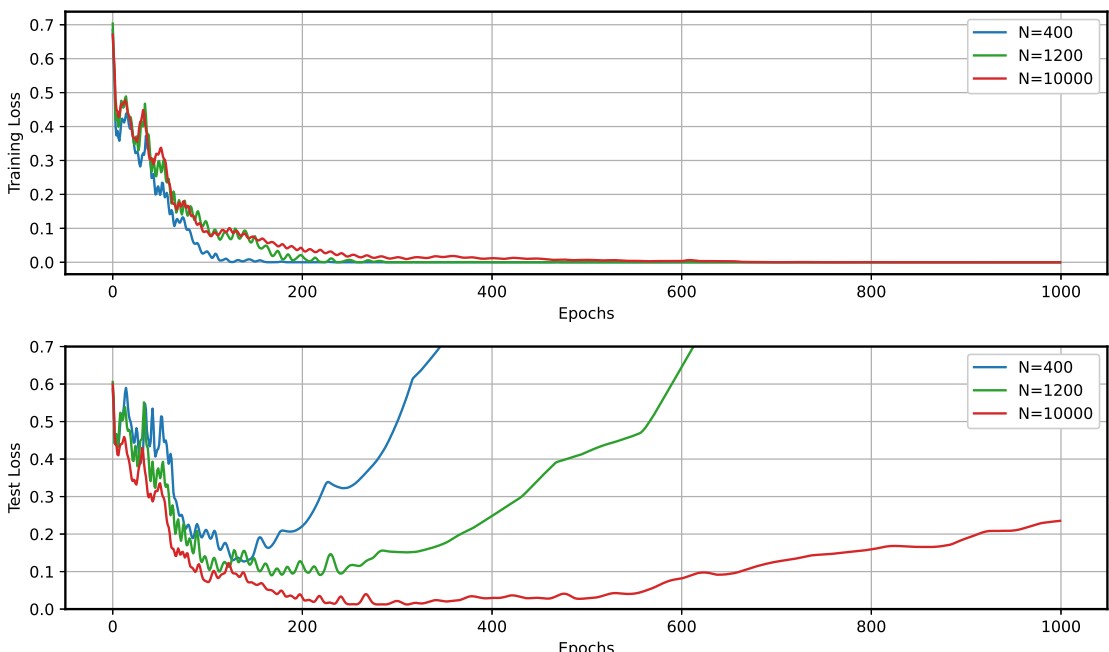

Figure 3: Evolution of training loss (top) and test loss(bottom) for each epoch of training for model dimension $d_m = 4096$.

Table 1: Lowest test loss and the epoch at which it was achieved for different values of $d_m$ and $N$.

| $d_m$ | $N$ | Lowest Test Loss | Epoch |
|-------|-------|------------------|-------|
| 64 | 400 | 0.1048 | 147 |
| 64 | 1200 | 0.0668 | 165 |
| 64 | 10000 | 0.0263 | 546 |
| 1024 | 400 | 0.1839 | 95 |
| 1024 | 1200 | 0.0760 | 271 |
| 1024 | 10000 | 0.0045 | 251 |
| 4096 | 400 | 0.1269 | 139 |
| 4096 | 1200 | 0.0899 | 169 |
| 4096 | 10000 | 0.0123 | 312 |

## B.2 Text Classification

We also perform similar experiments for text classification using the 20 Newsgroups dataset restricted to the categories *sci.med* and *sci.space*. Each text document is represented as a sequence of 40 tokens ($d_s = 40$), where each token corresponds to a 50-dimensional GloVe embedding ($d = 50$). This forms the input sequence for our shallow Transformer model. The main goal of these experiments is to demonstrate that the test loss of the trained Transformer model decreases as the number of training samples increases, i.e., for $N = 400$, $N = 1200$, and $N = 10000$. This decreasing trend in test loss is observed consistently across all model dimensions tested, namely $d_m = 64$ and $d_m = 1024$. The learning rate used in the experiments is 0.01, the optimization algorithm is batch gradient descent, and the loss function employed is the mean squared error (MSE) loss. The results are presented below, where each figure 4-5 shows the training loss and test loss trajectories of the Transformer model as training progresses over 2000 epochs.

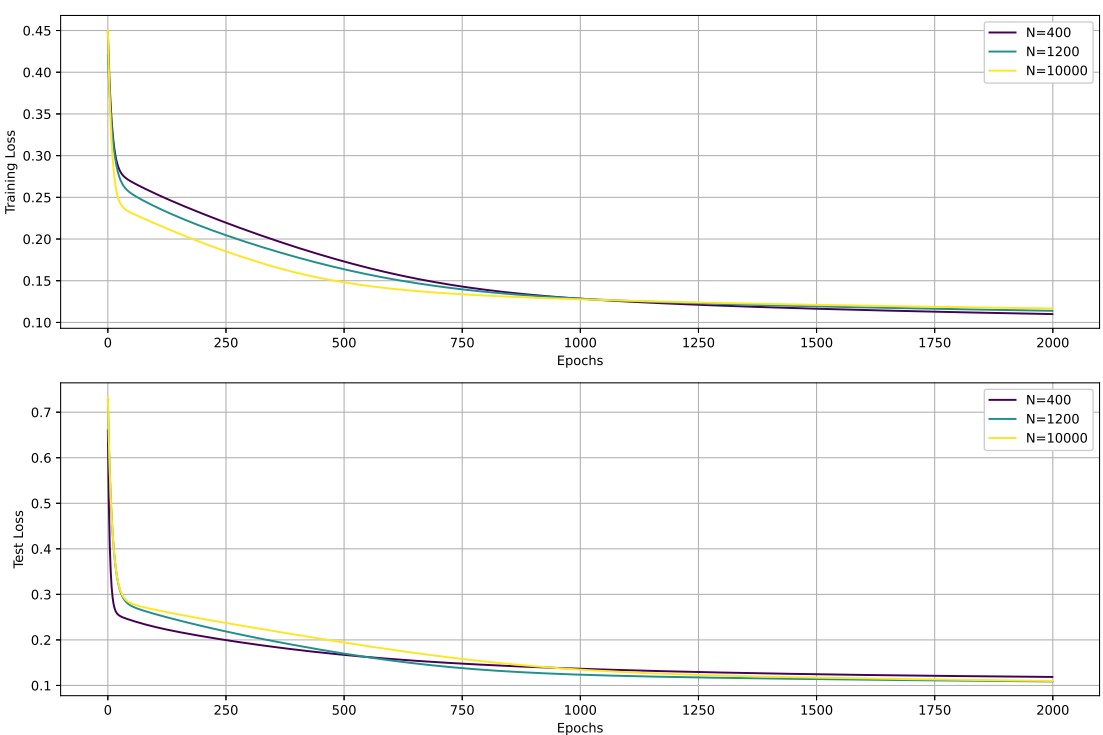

Figure 4: Evolution of training loss (top) and test loss(bottom) for each epoch of training for model dimension $d_m = 64$.

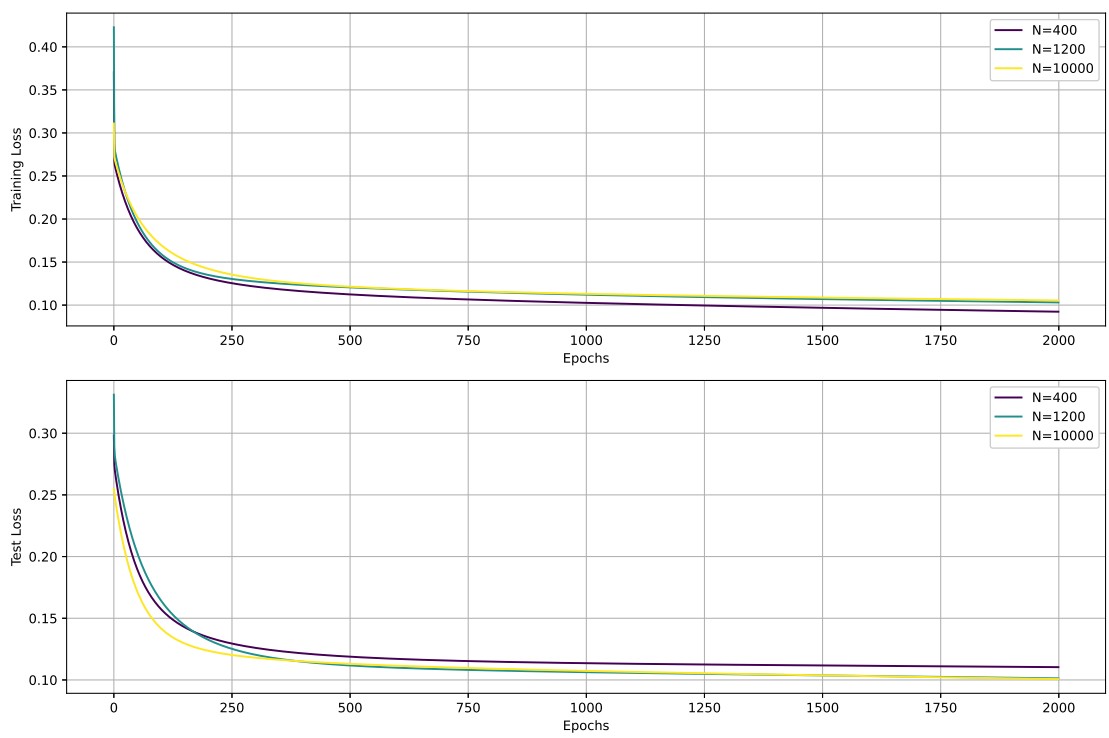

Figure 5: Evolution of training loss (top) and test loss(bottom) for each epoch of training for model dimension $d_m = 1024$.

