# OpenReview forum: "Generalization Bound for a Shallow Transformer Trained Using Gradient Descent"
_TMLR — Accepted by TMLR_

### Review · Reviewer_UayJ · 2025-08-28

**Summary Of Contributions:**

This paper studies the convergence of gradient on a restricted class of shallow transformers. The transformer architecture consists of the standard self-attention layer with a drastically simplified MLP layer with no learnable linear projections, and further, the class of models have the property that the model parameters do not deviate too much from initialization. The main result is an upper bound on the Rademacher process of this restricted class of transformers, and an upper bound on the empirical loss of 1-Lipschitz loss functions.

In my opinion, the results obtained in this paper are NOT very interesting. First, the assumptions on the transformer class are restrictive, in particular, 1. while the weights do not deviate from initialization is a key property in deep learning theory, but one should **prove** that gradient descent with proper step size indeed ensures the weights do not deviate too much, rather than assuming it directly, 2. in practice, MLP layer is quite important, this paper assumes the MLP layer with only ReLU activation, which largely restricts the complexity of the function class and the expressiveness of the model. Due to these simplifications, the proof is rather straightforward: one could simply assume initial bounds on the weights, which would directly lead to bounds on the weights at time $t$ due to the assumption, and the Rademacher process can then be upper bounded via Dudley's entropy integral. It adds little to no technical insights into proving the convergence of GD on transformers in general.

**Audience:**

Yes

**Audience Explanation:**

I think generalization for transformers is a very interesting problem, yet the restrictions of this paper are too limited to my opinion, this also limits its audience.

**Broader Impact Concerns:**

N/A.

**Claims And Evidence:**

No

**Claims Explanation:**

Proofs are provided, yet the justifications for the assumptions on the transformer class include the weights are close to the initialization and simplified MLP layer are quite weak. While it is known in deep learning theory that by tuning down the step size, the parameters do not deviate too much from initialization, which was the key to prove convergence for classical ReLU networks, authors should formally prove this is indeed the case when running GD with proper step size, rather than simply assuming it.

**Requested Changes:**

I believe in order for this paper to be accepted, it would significant strengthening of the results:

1. Instead of simplifying the MLP layer, authors should include the learnable linear projection of MLP;

2. Instead of assuming the weights do not deviate from initialization, authors should try to prove that gradient descent with proper step size indeed gives such condition. This would no doubt be much more difficult, authors should at least provide stronger and more convincing argument / evidence on **why** it makes sense to assume the weights will stay close to the initialization. As proven in the paper, in order for the generalization bound to hold one already requires the intermediate dimension $d_m\geq \Omega(N^3)$, such an overparametrization phenomenon is key in proving the convergence.

---

> ### Author Response · Authors · 2025-11-09
> **Scope of the paper and possible extensions**
>
> Requested change: Instead of simplifying the MLP layer, authors should include the learnable linear projection of MLP.
>
> Authors’ response: We thank the reviewer for this valuable suggestion. Our work builds directly upon the convergence analysis of shallow Transformers presented by Wu et al. (2024), which serves as the theoretical foundation for developing our generalization bound. In that work, the MLP layer is similarly simplified to enable a tractable convergence proof under the bounded-drift assumption.
> We fully agree that incorporating the learnable linear projection within the MLP layer represents an important extension. However, doing so would require establishing a new convergence theorem for this more general Transformer architecture—an ambitious theoretical undertaking that lies beyond the scope of the present study. Our primary objective is to leverage the existing convergence framework of Wu et al. (2024) to construct and analyze the generalization bound, rather than to rederive convergence guarantees for an extended model. We view the inclusion of a full learnable MLP projection as a promising direction for future work and have explicitly mentioned this in the revised manuscript.
>
>
>
>
> Requested change: Instead of assuming the weights do not deviate from initialization, authors should try to prove that gradient descent with proper step size indeed gives such condition. This would no doubt be much more difficult, authors should at least provide stronger and more convincing argument / evidence on why it makes sense to assume the weights will stay close to the initialization. As proven in the paper, in order for the generalization bound to hold one already requires the intermediate dimension , such an overparametrization phenomenon is key in proving the convergence.
>
> Authors’ response: We appreciate this insightful comment and fully acknowledge the importance of the bounded-drift assumption. Our current work does not aim to formally prove that gradient descent with an appropriately chosen step size ensures the weights remain close to their initialization. Instead, we adopt this assumption in line with a well-established body of work on the Neural Tangent Kernel (NTK) or lazy training regime, which characterizes the training dynamics of overparameterized neural networks under small learning rates.
> Empirical and theoretical results in this regime consistently demonstrate that the network operates in a near-linear region around initialization, leading to small parameter deviations during training. This phenomenon has been rigorously studied for wide neural networks (Jacot et al., NeurIPS 2018; Chizat & Bach, NeurIPS 2019) and extended to attention-based architectures, including Transformers (Yang, Tensor Programs II, 2020; Wu et al, 2024). Consequently, our assumption is grounded in these well-supported theoretical frameworks.
>
> In the revised manuscript (Section 4.5, Discussion), we have clarified this motivation and explicitly noted that, under sufficient overparameterization and small step sizes, gradient descent dynamics remain close to initialization—consistent with the NTK regime behavior observed in both theory and practice.

---

### Review · Reviewer_3jxZ · 2025-09-28

**Summary Of Contributions:**

The paper proposes a new norm-based generalization bound for a shallow Transformer trained with gradient descent. The approach builds on three interconnected components. First, the authors define a restricted hypothesis class of Transformer models whose parameters remain within a bounded neighborhood of their initialization at every training step. This assumption is motivated by the over-parameterized regime, where parameter drift is limited in practice. Second, they derive an upper bound on the Rademacher complexity of this class, employing covering number estimates to characterize the function space. Third, they upper bound the empirical loss for this class by invoking convergence guarantees for shallow Transformers trained with gradient descent, specifically relying on recent results about global convergence. Together, these steps yield a final theorem that provides an explicit upper bound on the true loss, which tightens sublinearly with the number of training samples for any model dimension. To provide empirical evidence, the authors conduct small-scale experiments on a binary subset of MNIST, showing that the test loss decreases as the number of training examples grows, and that this trend is consistent across model dimensions. The principal contribution lies in extending the methodology of directly bounding the true loss—previously demonstrated for fully connected networks—to the Transformer architecture.

The strengths of the paper are threefold. First, the work addresses an important gap in the theory of deep learning by providing a direct true-loss bound for Transformers, which had not previously been studied in this form. Second, the proofs are thorough and mathematically careful, with detailed derivations of the Rademacher complexity bound and a clear connection to convergence results. Third, the structure of the argument is well-motivated: the authors justify their assumptions, systematically develop the lemmas, and integrate them into a coherent theorem that contributes meaningfully to our understanding of Transformer generalization.

Despite these strengths, the paper also suffers from weaknesses. The bounded-drift assumption on parameters is very restrictive, and while the authors argue that this reflects over-parameterized training, the conditions are not carefully validated in practice. The empirical section is underdeveloped: using only two-class MNIST, without comparisons to alternative bounds or to real-world Transformer training, makes the evidence weak relative to the theoretical ambition. Finally, the contribution is somewhat narrow. The focus is restricted to shallow, single-head Transformers, and while the authors suggest extensions to deeper and more general models, no concrete results are provided, which limits the impact.

**Additional Comments:**

Overall, this paper makes a meaningful but narrow contribution to the theoretical understanding of Transformers. Its strength lies in extending direct true-loss bounds to a new architecture with rigorous mathematical treatment. Its main weakness lies in the limited empirical evidence and the restrictive assumptions, which limit practical significance. With stronger experiments and clearer positioning within the broader literature, the paper would merit acceptance. As it stands, it is a borderline case: strong on theory but weak on applicability.

**Audience:**

Yes

**Audience Explanation:**

Yes, the theoretical community studying generalization in deep learning, and in particular those focusing on Transformers, will find the results of interest. The extension of direct true-loss bounds to this architecture fills a gap in the literature and contributes to the theoretical toolbox available for analyzing large models. However, the readership interested in applied or empirical aspects of Transformers may be less engaged, since the results are highly stylized and the experiments are too modest to carry practical significance.

**Broader Impact Concerns:**

There are no major ethical concerns raised by this work. The paper is entirely theoretical and does not involve sensitive data, applications, or downstream deployment. The broader impact statement should, however, acknowledge explicitly that the results are limited to shallow and constrained Transformer models, and thus their implications for the safety or reliability of large-scale Transformer deployment are speculative.

**Claims And Evidence:**

Yes

**Claims Explanation:**

Yes, the claims are supported in the sense that the theoretical derivations are correct and internally consistent. The use of covering numbers and Rademacher complexity is rigorous, and the application of the shallow Transformer convergence theorem is appropriate. However, the evidence is not fully convincing with respect to practical relevance. The experiments are too limited to demonstrate that the bound meaningfully predicts generalization in real settings. The authors overstate the empirical side of their contribution, since the experiments only verify a trivial monotonic relationship between sample size and test loss. Thus, while the theoretical claims are well supported, the practical claims are weak.

**Requested Changes:**

A critical change is the need for stronger empirical validation. At minimum, the experiments should move beyond two-class MNIST and include synthetic or toy sequence modeling tasks that more closely resemble the intended domain of Transformers. Without this, the experimental section adds little value. A second critical change is to clarify the scope and limitations of the main theorem, especially regarding the bounded-drift assumption and the shallow architecture. The current text suggests broader implications than are warranted. A third critical improvement would be to situate the contribution more carefully against existing generalization bounds for Transformers, beyond the short discussion of Edelman et al. (2021) and Trauger & Tewari (2024). Comparisons to other frameworks, such as PAC-Bayes or stability-based approaches, would help contextualize the novelty. Non-critical but useful improvements include polishing the presentation of proofs for readability, especially in the long lemma sections, and expanding the discussion section to include more explicit insights about what the bound implies in practice.

---

> ### Author Response · Authors · 2025-11-09
> **Added discussion section and experiments**
>
> Requested change: A critical change is the need for stronger empirical validation. At minimum, the experiments should move beyond two-class MNIST and include synthetic or toy sequence modeling tasks that more closely resemble the intended domain of Transformers. Without this, the experimental section adds little value.
>
> Authors’ response: We thank the reviewer for this valuable suggestion. In response, we have added new experiments on sequential data in Section B.2 to better align with the intended application domain of Transformers. These additional experiments complement the MNIST results and provide empirical support for our theoretical findings in a more representative setting.
>
>
>
> Requested change: A second critical change is to clarify the scope and limitations of the main theorem, especially regarding the bounded-drift assumption and the shallow architecture. The current text suggests broader implications than are warranted.
>
> Authors’ response: We appreciate this important feedback and have revised the manuscript to clearly delineate the scope and limitations of the main theorem. Specifically, in Section 4.5 (Discussion), we now explicitly state that the bounded-drift assumption corresponds to the lazy training regime, where model parameters remain close to their initialization—an assumption most valid for wide, shallow Transformers trained with small learning rates. We further clarify that our generalization bound applies exclusively to single-layer (shallow) architectures and that extending the analysis to deeper or non-lazy regimes would require additional theoretical developments. These revisions ensure that our claims are appropriately contextualized within the theoretical scope of the work.
>
>
> Requested change: A third critical improvement would be to situate the contribution more carefully against existing generalization bounds for Transformers, beyond the short discussion of Edelman et al. (2021) and Trauger & Tewari (2024). Comparisons to other frameworks, such as PAC-Bayes or stability-based approaches, would help contextualize the novelty.
>
> Authors’ response: We have incorporated a more comprehensive discussion of related frameworks. In the Related Work section, we now include comparisons to PAC-Bayes and stability-based approaches to better situate our contribution within the broader literature. Additionally, in Section 4.5 (Discussion), we have added dedicated paragraphs comparing our results with existing norm-based bounds, as well as with PAC-Bayes and stability-based generalization frameworks. These additions provide clearer context for the novelty and distinct analytical perspective of our work.
>
>
> Requested change: Non-critical but useful improvements include polishing the presentation of proofs for readability, especially in the long lemma sections, and expanding the discussion section to include more explicit insights about what the bound implies in practice.
>
> Authors’ response: A discussion section has been added in section 4.5 and introductory paragraphs have been added to each results section to contextualize the results from each lemma.We appreciate this helpful recommendation. In the revised manuscript, we have refined the presentation of proofs to improve clarity and readability, particularly within the longer lemma sections. Furthermore, we have expanded the Discussion (Section 4.5) to provide more explicit insights into the practical implications of the derived bounds. Introductory paragraphs have also been added to each results subsection to better contextualize the purpose and interpretation of each lemma.

---

### Review · Reviewer_aNGY · 2025-10-12

**Summary Of Contributions:**

The main contribution of this paper is establishing the first direct generalization bound for shallow Transformers by integrating training dynamics analysis with Rademacher complexity, achieving an $O(\sqrt{P/N})$ convergence rate.

**Additional Comments:**

NA

**Audience:**

Yes

**Audience Explanation:**

Transformer generalization theory is a timely topic, and the methodology of combining training dynamics with complexity analysis is novel. Despite the technical errors, the corrected work would provide valuable theoretical tools for understanding Transformer generalization.

**Claims And Evidence:**

No

**Claims Explanation:**

The paper contains two technical errors, which compromise the rigor and numerical accuracy of the results:
- In Lemma 1, the reduction from $\mathcal{F}$ to $\mathcal{G}$ incorrectly equates matrix "full rank" with "statistical independence" of rows, deriving the equality $\mathcal{R}_S(\mathcal{F})=\mathcal{R}_S(\mathcal{G})$. However, full rank only implies linear independence; subadditivity of supremum should yield an inequality instead.
- In Lemma 4, the second term should be $\frac{\alpha^3 \eta_O}{32\rho^2\hat{z}\sqrt{N}}$, but the paper incorrectly writes $\rho^4$.

**Requested Changes:**

1. Remove the independence claim; apply subadditivity to obtain $\mathcal{R}_S(\mathcal{F}) \leq d_s \cdot \mathcal{R}_S(\mathcal{G})$; adjust sequence length dependence accordingly.

2. Correct the exponent from $\rho^4$ to $\rho^2$ in the second term; verify the derivation chain; update Theorem 5 and all dependent results.

---

> ### Author Response · Authors · 2025-11-09
> **Corrections in the proof**
>
> Requested change: Remove the independence claim; apply subadditivity to obtain $\mathcal{R}_S(\mathcal{F}) \leq d_s \cdot \mathcal{R}_S(\mathcal{G})$; adjust sequence length dependence accordingly.
>
> Authors’ response: We thank the reviewer for this insightful observation. The proof of Lemma 1 has been revised to remove the incorrect implication between full rank and statistical independence. We now explicitly apply the subadditivity property of the supremum, leading to the inequality form of the bound. Furthermore, we clarify that $\mathcal{R}_S(\mathcal{G}_R^{\boldsymbol{\theta}^0})$ is defined with a normalization factor of $\tfrac{1}{N}$, resulting in the final inequality $\mathcal{R}_S(\mathcal{F}_R^{\boldsymbol{\theta}^0}) \le \mathcal{R}_S(\mathcal{G}_R^{\boldsymbol{\theta}^0})$. If the normalization were instead $\tfrac{1}{N d_s}$, the resulting bound would be $\mathcal{R}_S(\mathcal{F}_R^{\boldsymbol{\theta}^0}) \le d_s \cdot \mathcal{R}_S(\mathcal{G}_R^{\boldsymbol{\theta}^0})$.
>
> Requested change: Correct the exponent from $\rho^4$ to $\rho^2$ in the second term; verify the derivation chain; update Theorem 5 and all dependent results.
>
> Authors’ response: We have corrected the exponent of $\rho$ from $\rho^4$ to $\rho^2$ in Lemma 4 and its proof, reflecting the proper reduction that occurs after taking the square root in the loss bound derivation. This correction has been consistently propagated to Theorem 5 and all subsequent dependent results. The derivation chain has been carefully reverified to ensure consistency throughout.